# Classification of Nanomaterials and the Effect of Graphene Oxide (GO) and Recently Developed Nanoparticles on the Ultrafiltration Membrane and Their Applications: A Review

**DOI:** 10.3390/membranes12111043

**Published:** 2022-10-26

**Authors:** Raghad M. Al-Maliki, Qusay F. Alsalhy, Sama Al-Jubouri, Issam K. Salih, Adnan A. AbdulRazak, Mohammed Ahmed Shehab, Zoltán Németh, Klara Hernadi

**Affiliations:** 1Membrane Technology Research Unit, Department of Chemical Engineering, University of Technology-Iraq, Alsinaa Street 52, Baghdad 10066, Iraq; 2Department of Chemical Engineering, College of Engineering, University of Baghdad, Aljadria, Baghdad 10071, Iraq; 3Department of Chemical Engineering and Petroleum Industries, Al-Mustaqbal University College, Babil 51001, Iraq; 4Faculty of Materials and Chemical Engineering, University of Miskolc, H-3515 Miskolc, Hungary; 5Polymers and Petrochemicals Engineering Department, Basrah University for Oil and Gas, Basrah 61004, Iraq; 6Advanced Materials and Intelligent Technologies Higher Education and Industrial Cooperation Centre, University of Miskolc, H-3515 Miskolc, Hungary; 7Institute of Physical Metallurgy, Metal Forming and Nanotechnology, University of Miskolc, H-3515 Miskolc-Egyetemváros, Hungary

**Keywords:** mixed matrix membrane, nanoparticles, graphene oxide (GO), tungsten oxide (WO_x_), polyethersulfone (PES), polyphenylsulfone (PPSU), polyvinyl chloride (PVC), ultrafiltration membrane, environments, wastewater treatment

## Abstract

The emergence of mixed matrix membranes (MMMs) or nanocomposite membranes embedded with inorganic nanoparticles (NPs) has opened up a possibility for developing different polymeric membranes with improved physicochemical properties, mechanical properties and performance for resolving environmental and energy-effective water purification. This paper presents an overview of the effects of different hydrophilic nanomaterials, including mineral nanomaterials (e.g., silicon dioxide (SiO_2_) and zeolite), metals oxide (e.g., copper oxide (CuO), zirconium dioxide (ZrO_2_), zinc oxide (ZnO), antimony tin oxide (ATO), iron (III) oxide (Fe2O3) and tungsten oxide (WO_X_)), two-dimensional transition (e.g., MXene), metal–organic framework (MOFs), covalent organic frameworks (COFs) and carbon-based nanomaterials (such as carbon nanotubes and graphene oxide (GO)). The influence of these nanoparticles on the surface and structural changes in the membrane is thoroughly discussed, in addition to the performance efficiency and antifouling resistance of the developed membranes. Recently, GO has shown a considerable capacity in wastewater treatment. This is due to its nanometer-sized holes, ultrathin layer and light and sturdy nature. Therefore, we discuss the effect of the addition of hydrophilic GO in neat form or hyper with other nanoparticles on the properties of different polymeric membranes. A hybrid composite of various NPs has a distinctive style and high-quality products can be designed to allow membrane technology to grow and develop. Hybrid composite NPs could be used on a large scale in the future due to their superior mechanical qualities. A summary and future prospects are offered based on the current discoveries in the field of mixed matrix membranes. This review presents the current progress of mixed matrix membranes, the challenges that affect membrane performance and recent applications for wastewater treatment systems.

## 1. Introduction

With significant population growth and the impacts of climate change, water availability poses a serious challenge to global water security. Therefore, appropriate solutions have to be found to meet the use and supply of water over time while maintaining the quality of the water [1]. Over the years, various technologies have been studied to supply clean water sources. Sewage treatment and recycling, industrial wastewater treatment and seawater desalination are all used for supplying water [2]. Many wastewater treatment techniques have been applied to eliminate the barriers between water supply and demand. For example, sedimentation [3] and flotation [4], which are used for primary treatment, can be employed for the removal of large particles, microbiological entities and oil spills; however, they are less efficient for microbial removal. Additionally, for low concentrations of pollutants, adsorption [5,6] and ion exchange [7,8] have been considered for organic and metal ions removal. Relative to classic treatment technologies, membrane technologies, which were initially developed for water treatment, have been considered for their effective separation, high capacity, being inexpensive, reliable and simple, offering clean separation methods being and environmentally better than other methods [9,10].

As a result, membrane separation techniques have been successfully applied in the treatment of discharges from different industries such as the metallurgical industry, leather industry, pharmaceutical industry, food processing, petrochemical industry, oil and gas industry and desalination. [11]. Membrane technology has been widely utilized to remove contaminants, such as bacteria, oil and dissolved organic or inorganic species. An efficient separation and purification highly depend on the separation efficiency of these membranes in the industry. In addition, these membranes contribute to the popularity of their use in the separation process by having a low power consumption and small footprint and being easy to handle [12,13].

Using ultrafiltration (UF) membranes as a part of membrane separation technology are a clean, efficient, and desirable method that successfully filters suspended particles, microorganisms and organic compounds. UF membranes play an important role in the production of clean water because they are more economic and efficient treatment processes for a wide range of contaminants than nanofiltration (NF) [14,15] and reverse osmosis (RO) [16,17]. UF is clean, safe, simple to use and highly efficient in separating proteins, bacteria, viruses and turbidity. The majority of water treatment studies have focused on separation efficiency and fouling resistance [2]. UF membranes can be fabricated by the phase inversion method using different polymeric materials, such as polyethersulfone PES [18,19,20,21,22,23,24,25,26], polyvinylidene fluoride (PVDF) [27,28,29,30], polyacrylonitrile (PAN) [31,32,33], bromomethylated polyphenylene oxide (BPPO) [34,35] and PVC [11,36,37]. Although commercial polymer materials are available, their low permeation flux and fouling issues limit their use [30]. The final membrane texture is influenced by its composition (concentration, solvent and organic or inorganic additives) [38]. In addition, the temperature of the polymer solution, non-solvent or non-solvent mixture, and coagulation bath or the environment influence the produced membrane [39] because it influences the membrane’s chemical, thermal and mechanical properties [40].

The main difficulty faced in membrane technology is membrane fouling. Membrane fouling has been linked to hydrophobicity in several studies [39]. This is because fouling affects the membrane flux permanently or temporarily by reducing the flow across the membrane rapidly, reducing the target compound rejection and the membrane’s lifetime, thus increasing operating costs [30,41]. Fouling affects a polymeric membrane through the interactions between the polymeric membrane’s surface charges and foulants, which can be organic, inorganic or biological forms [30]. The adsorption of organic compounds on the membrane surface is commonly responsible for fouling; however, other types of fouling can also occur, such as bio fouling and scaling. Therefore, the characteristics of the membrane and the filtration method, cross-flow or dead-flow filtration, affect the generation of fouling. The membrane application is critically threatened by fouling formation and the utilization of membranes in industrial applications is severely limited unless this problem is solved [41]. Therefore, improving the hydrophilicity of the polymeric membrane is an important task because it can improve the separation efficiency and antifouling performance of UF membranes [30].

Adding hydrophilic NPs to the polymeric membranes reduces their contact angle and increases their hydrophilicity, increasing the separation performance of pollutants. However, adding hydrophilic polymers, such as poly(ethylene glycol) (PEG), poly(vinyl pyrrolidone) (PVP), and cellulose acetate phthalate, to the polymeric membranes works as pore formers, which increases the hydrophilicity of the membranes at the expense of reducing the separation performance [42].

Organic fouling as well as bio-fouling can be noticed in the indirect production of drinking water from supplementary wastewaters. This can be caused by organic matter, which can come easily from drinking water sources, synthesized from disinfection byproducts or rather domestic use, and biomass [43]. There are four primary mechanisms of fouling, as shown in Figure 1a–d, which are intermediate blocking, complete blocking, standard blocking and cake formation respectively. Intermediate blocking happens when particles are larger than the usual particle size, but only some of the particles block some of the pores, whereas the remaining particles are deposited on top of the surface. In complete blocking, the particles are larger than the pores of the membrane, so they prevent the flow because they block the pores. However, when the particles are smaller than the pores of the membrane, they accumulate on the wall of the pores, so the flow is reduced; this type of fouling is called standard blocking. The most complicated type of the fouling is cake formation, which takes place when the particles of the fouling materials accumulate on the surface of the membrane because of their size being larger than the pore size [43,44,45].

A technique was employed by Grace and Hermia to differentiate among different fouling mechanisms. Hermia’s model’s parameters have physical significance and enable the better estimation of the dominant mechanisms of reduction in permeability. Additionally, every mechanism has a formula, as seen in Table 1 [46]:

Maiti et al. studied the membrane fouling mechanism in a batch dead-end ultrafiltration cell with a 100 kDa PES membrane at a stable operating pressure of 1.36 bar. They plotted the UF’s ln(J) against time. Across all tests, the cake filtering model outperformed the other models in terms of linearity, specifically at lower pH levels (R2 > 0.98). In these tests, the data show that full pore plugging would have been the lowest likely mechanism of flow reduction [46].

The hydrophobicity of polymeric membranes can be reduced in various ways. The methods that can generate more hydrophilic surfaces are: (1) graft polymerization, a chemically attached graft polymerization method; (2) plasma treatment, which provides various functionalities on the membrane surface; and (3) adding hydrophilic components that are physically adsorbed on the membrane surface [39]. The chemical methods used for reducing the hydrophobicity of polymeric membranes include photo-induced grafting, gamma ray and electron beam-induced grafting, plasma treatment and plasma-induced grafting, thermal-induced grafting and immobilization, and surface-initiated atom transfer radical polymerization, which have all been used to modify polymeric membranes. Additionally, many new polymerization techniques, such as the reversible addition of fragmentation chain transfer (RAFT) polymerization and click chemistry approaches, have been developed to modify polymeric membranes [39].

Nanotechnology is a well-studied approach for producing antifouling membranes with great separation performance. GO [37,47,48], MOFs [49,50], zeolites [25,51], Al_2_O_3_ [52], carbon nanotubes (CNTs) [27,45,46,48,53,54], (SiO_2_) [55], ZnO [56], mesoporous MCM-41 [26], WO2.89 [57] and other nanoparticles (NPs) have been employed as nanofillers in the UF mixed matrix membranes (MMMs). In terms of flow, rejection and fouling resistance, they perform quite well. GO and GO-based materials have been identified as viable nanofillers for improving membrane fouling resistance and separation performance because of their excellent chemical stability, mechanical strength, ease of accessibility, ultrathin layer, nanometer-sized holes, and lightweight and sturdy nature [2].

In recent years, researchers have created 2D materials such as graphene/graphene oxide (GO), COFs and MOFs. In addition to these materials, 2D MXene materials have also attracted the attention of researchers because it has only been recently added for membrane fabrication and has unique chemical properties [58]. Because of their extendable layered structure as well as remarkable mass transfer channel, two-dimensional materials have recently demonstrated broad application prospects for the fabrication of high-performance membranes [59,60]. Recent research has shown that inorganic MXene materials can not only enhance the antifouling of the membranes but also improve removal efficiency [61].

Mixed matrix membranes (MMMs) have attracted the attention of researchers worldwide, and the number of annual publications returned by the Google Scholar and ScienceDirect databases has grown continuously, doubling dozens of times from 2010 to 2022, and still continuing to increase significantly. This indicates the great impact of MMMs on the performance of membranes, which in turn prompted us to focus, in this review, on the study of the effect of different NPs on physicochemical and mechanical properties and performance for resolving environmental and energy-effective water purification. More specifically, this review focuses on the effect of GO and hyper with other NPs as well as recently developed nanoparticles such as MXenes, MOFs and COFs due to their excellent properties on membrane characteristics to enhance the removal efficiency of pollutants from wastewater.

## 2. Mixed Matrix Membranes (MMM)

The structural properties and performance of polymeric membranes have been improved by various methods, such as the addition of water-soluble compounds and NPs as well as by free-radical graft copolymerization (chemical modification) [30]. Embedding polymeric membranes with inorganic fillers, such as SiO_2_, ZnO and TiO_2_, creates a special type of texture matrix, called a mixed matrix membrane (MMM). The main purpose of creating such a mixture is to incorporate the favorable properties of inorganic materials to enhance the total effectiveness of the produced membrane matrix. MMMs have attracted more interest than polymeric membranes because of their high ability to remove particular pollutants and their antifouling properties [62].

### 2.1. Classification of Nanomaterials

Hydrophilic NPs have been employed as fillers in many researches to improve the hydrophilicity and antifouling performance of UF membranes [30]. Membrane separation has developed in recent years due to the use of nanomaterials in the synthesis of membranes because of the ability of inorganic fillers of improving membrane characteristics, especially when NP additives are used, i.e., additives ranging, in the nanoscale, from 1 to 100 nm [63]. Table 2 shows the studies encompassing the preparation of MMMs with different types of NPs and different approaches. All additives have cons and pros when used, so this is why all these researches aim to solve the membrane’s problem as well as embrace the benefits and good properties of NPs.

#### 2.1.1. Mineral Nanomaterials

##### SiO_2_

The fabrication of MMMs using SiO_2_ particles has shown improvement in the hydrophilicity of composite membranes because of the hydrophilic nature of these particles. However, they make no discernible effect on the cross-sectional structures. The macrostructure was unaffected by the added quantity of SiO_2_ particles, but they changed the inner pore size of the membrane [64]. This means that SiO_2_ particles can be useful for the fabrication of UF membranes; hence, many research articles used these particles in their works [62,65]. Al-Araji et al. developed MMMs by adding SiO_2_ NPs to PES UF membranes and incorporated SiO_2_ in a PEI polymer to increase the dispersion of SiO_2_ in the PES polymer. Al-Araji et al. separated Congo red dye and reactive black dye and obtained the following removal percentages, 94.1% and 90.5%, respectively, with 39.7 L/m^2^ h and 43.3 L/m^2^ h for the permeation flux of Congo red dye and reactive black dye, respectively. Therefore, the inclusion of SiO_2_ improved the hydrophilic nature of the membrane that is related to hydroxyl as well as amine functional groups in SiO_2_ nanoparticles, which were attributed to their hydrophilicity [66].

##### Zeolite

Since zeolite contains cations, it has a considerable advantage as an ion-exchange agent (sodium, calcium or potassium). In solutions of cadmium, lead, zinc and manganese, such cations can be exchanged for other positive cations. Therefore, MMMs have a strong ability, when mixed with zeolite, to remove heavy metals because of the effects that zeolite provides, such as selectivity and highly surface area. Alfalahy et al. successfully integrated a NaX zeolite powder at various percentages in the PES membrane to improve the efficiency of the ultrafiltration separation process for Pb(II) ions in an aqueous solution. As the zeolite load increased over 0.6 wt%, the membrane became denser and the pore volume decreased, water permeability increased and 97% of Pb(II) ions were removed because of the hydrophilic nature of the NaX zeolite [25]. Multiple techniques, such as ion exchange/sorption, evaporation, chemical precipitation and membrane separation, have been developed to improve Cesium’s selectivity and remove it from liquid nuclear waste because of its dissolved nature at low quantities in wastewaters, poisonous nature and higher danger of internal radiation exposure when compared to other radioactive elements. Abbas et al. explained that NaY–Zeolite is useful to remove 137 Cs from wastewater by incorporating zeolite nanoparticles into a PES polymeric membrane. The membrane’s ion exchange efficiency is thus improved. This is because the PES/zeolite membrane contains a negative charge produced by Si–O as well as Al–O. The negative charge attracts cations such as Cs+, which become attached to the membrane. Abbas et al. obtained a 99.2% removal of cesium, obtaining 0.15 wt.% zeolite with a 97.8 (L m−2 h−1) permeate flux. The hydrophilicity of the membrane is thus significantly enhanced. This is related to the transition from a hydrophobic to a hydrophilic effect. These MMMs are considered environmentally friendly methods of treating nuclear waste from multiple resources [67].

#### 2.1.2. Metal Oxides

##### CuO

The inclusion of CuO NPs in the PES membranes was intended to improve the permeability and antifouling properties of the membrane. The antifouling performance of a composite membrane containing 0.1 wt.% CuO was the best, with a flux recovery ratio (FRR) of 60% and a BSA retention value of more than 97% [23]. Nasrollahi et al. [23]. reported that inclusion CuO NPs to the membranes’ precursor reduced the water contact angle due to the hydroxyl groups of CuO, resulting in an increase in hydrophilicity and water flux. Additionally, a significant number of research using CuO NPs has been presented to date [68,69]. Moreover, CuO/ZnO is an effective additive in PES membranes, according to the literature [18]. It was added with 0.2 wt.% and its addition produced a BSA rejection of more than 98% and the antifouling properties and the permeability of the prepared composite membranes were noticeably improved.

##### ZrO_2_

According to Pang et al. [70], the strength of the membrane containing zirconium oxide (ZrO_2_) NPs was higher than that of a pristine membrane, obtaining a higher membrane permeability and improved membrane surface features. Membranes containing ZrO_2_ are chemically more stable than those containing TiO_2_ and alumina (Al_2_O_3_) NPs, and thus they are better suited for liquid phase applications in severe environments [70]. Wen et al. [71] showed that the presence of ZrO_2_ NPs in the precursor prevents aggregation during the drying and sintering steps accompanied by the fabrication of a tight UF membrane. Therefore, the applications of ZrO_2_ in UF are increasing day by day [72,73].

##### ZnO

ZnO is a common low-price metal oxide that has recently been utilized instead of TiO_2_ [63]. ZnO has excellent physical and chemical properties: high photosensitivity, thermal stability, antibacterial action and optical absorption in the UV area [19]. ZnO can produce membranes with high hydrophilic character with high porosity and flux, while moderate rejection is also observed. The antifouling properties of the membrane prepared with ZnO were superior when acetone was used as a solvent because it reduced fouling by forming a tight skin layer. Since a combination of ZnO/acetone has a moderate antifouling performance and excellent membranes can be synthesized with these two materials with high fouling resistance [20]. A study conducted by Rabiee et al. [74] reported that adding 3 wt% of ZnO to the membrane precursor improved the water flow, flux recovery and BSA rejection, but adding above this load produced a reverse impact on water flow, flux recovery and BSA rejection. Another study performed by Alsalhy et al. [75] reported that increasing ZnO concentration in the casting solution of PPSU led to changes in the membrane’s structure from a finger-like structure in the layer near the polyester support to a completely sponge-like structure. Additionally, the hydrophilicity of the membrane was improved and the mean pore size reduced as the concentration of ZnO NPs in the PPSU casting solution increased. The permeability of pure water through PPSU membranes improved as the ZnO concentration increased to 0.025 wt.%; however, adding 0.03 wt.% of ZnO reduced pure water permeability. The addition of ZnO NPs to the casting solution had no effect on the performance of the solute separation of the PPSU membranes [75].

##### ATO

The use of ATO NPs in various areas of research and technology is rapidly expanding, owing to their high electrical conductivity and low synthesis cost. According to Khorshidi et al., ATO NPs were synthesized using a modified solvothermal technique; then, they were homogeneously scattered in the casting solution to make PES-ATO nanocomposite membranes through the phase inversion method. Because of their larger surface hydrophilicity and smaller pore size, Khorshidi et al. found modified membranes with ATO NPs have enhanced antifouling tendency. The modified membranes were also found to outperform the pristine PES membranes in the areas of organic matter separation efficiency and water flux recovery [76].

##### Fe_2_O_3_

Liu et al. [77] found the flux was improved by 66% and reach 138 L m^−2^ h^−1^ with a decrease of 22% in the contact angle when embedded 1.0% Fe_2_O_3_ in a PVC membrane. According to Liu et al., Fe_2_O_3_ modification improved the PVC membrane’s average flux and antifouling performance, because the overall irreversible backwashing fouling proportion of the 1.0% Fe_2_O_3_ embedded in PVC membranes was 0.27. The membrane of PVC membranes with 1.0% Fe_2_O_3_ had fewer foulants than unmodified PVC membrane.

##### Tungsten Oxide (WO_x_)

WOx is a transition metal oxide with a wide range of applications, especially in re-cent years due to its electrochromic (EC) effect. Nanostructured WOx is exceptionally versatile and offers unique characteristics. It has become one of the most investigated metal oxides impacting many research fields ranging from condensed-matter physics to solid-state chemistry [78]. Table 2 shows the properties of MMMs prepared with many types of NPs that have been used in UF all over the world. It can be seen from Table 2 that different transition metal oxides have been used as inorganic additives to prepare MMMs, either individually or coupled with other transition metals oxide or GO, multi-walled carbon nanotubes (MWCNT), etc. Additionally, it shows that all prepared MMMs had low contact angles, which explained the high rejection obtained for a chosen pollutant.

#### 2.1.3. Two-Dimensional Transition Metal

Two-dimensional materials have been studied in depth in recent years in various separation applications, such as GO, as detailed in Section 2.1.6, graphene [79], transition metal dichalcogenides (TMDS) [80], boron nitride (BN) [81] and metal carbides and nitrides (MXenes).

In situ reduction technology was used to create Ni@MXene magnetic particles, which were then attached by external magnetic field to the top surface of the PES membrane. Huang et al. found that the CR solution and colored emulsion demonstrated excellent decolonization ability. Furthermore, the antifouling mechanism might be demonstrated by the fact that the interaction between the Ni@MXene membrane and pollutants is relatively resistant compared to the pure membrane [61], while Shen et al. [82] prepared a membrane by embedding MXene throughout the phase inversion by diffusing them in a coagulation bath. When compared to the pure PSF membrane, all composite polymeric membranes demonstrated significant improvements in water flux as well as BSA rejection. The improved antifouling feature is due to enhanced surface smoothness, increased hydrophilicity and the MXene nanosheets’ more negative zeta potential. Because of these significant enhancements [82], MXene has obtained considerable attention for its ability to remove heavy metal ions from the wastewater, such as Cu^2+^, Cd^2+^ and Cr^6+^. Yang et al. [83] used different concentrations of Fe_3_O_4_NPs in the 2D MXene lamellar structure to create composite NF membranes. The removal ratios of heavy metals were 63.2% for Cu^2+^, 64.1% for Cd^2+^ and 70.2% for Cr^6+^ [83]. One of the modification methods is composite nanomaterials for membrane fabrication. Multiple nanomaterials with hydrophilic groups were added into the membranes to improve hydrophilicity. Huang et al. studied the separation and anti-fouling ability, preparing a TiO_2_@MXene composite and introducing it into the PES polymeric membrane. There were higher fluxes obtained with 1 g/L BSA solution (756.8 L/m^2^·h^-1^) and BSA rejection reached 70% [59]. MXene was used also for the removal of oil and dyes from wastewater. For example, Ajibade et al. studied the preparation of a MXene/O-MWCNT@PAN mixed matrix membrane for ultrafiltration membrane applications. The engagement of O-MWCNT within MXene nanosheets inside the membrane led to the quick passing of water molecules, while preventing the passage of oil droplets. Throughout the operational period, the rejection rates for oil and dye were about 97% and 99%, respectively. This describes the anti-swelling characteristic of the composite membrane’s 3D MXene/O-MWCNT nanoparticles [84].

#### 2.1.4. Metal–Organic Frameworks (MOFs)

MOFs that are composed of metallic ions and organic ligands have received a considerable amount of attention in the last decade due to their exceptional high surface area and porosity, functionalization capability, affinity for specific molecules, tunable chemical composition and flexible structure. In comparison to traditional inorganic materials with ‘rigid’ structures, the organic nature of MOFs has a better compatibility with soft polymer matrices. These benefits may result in MOF/polymer blended membranes with increased permeability and stable solute rejection [85,86,87]. Ma et al. found that applying MOF@GO composites as fillers to ultrafiltration membranes is a highly effective and promising technique for producing advanced water purification membranes. The hydrophilicity as well as water purification efficiency of UiO-66@GO/PES membranes were significantly improved after combining UiO-66@GO into the PES membrane matrix. The pure water flux of the composite membrane was increased by 351% when compared to the pristine PES membrane. The antifouling measurements demonstrate an excellent antifouling performance [88].

#### 2.1.5. Covalent Organic Frameworks (COFs)

COFs have also recently received considerable attention in the fields of energy storage, catalysts and separation. COFs, a new generation of crystalline natural porous material, consist of H, N, C, O, B, as well as other light atoms and have unique properties, including inevitable porous structure, good porous aperture and active functional groups [89]. Additionally, it also a promising material for membrane modification in treating wastewater and heavy metal recovery processes as Xu et al. studied COFs/PVDF in lead removal [90]. Zhang et al. improved loading COFs materials under the best conditions, showing that it could effectively have efficient results for dye separation. Furthermore, the composed COF-based membrane passed the testing (which lasted 30 h) and proved that it has great long-term stability and is also extremely dependable under highly acid/base conditions. As a result, the synthesized membrane seems to have the advantages of large-scale production [91].

**Table 2 membranes-12-01043-t002:** Characteristics of UF MMMs prepared with different polymers and additives.

Type of Polymer	Type and Composition of NPs	%Porosity	Pore Size (nm)	Contact Angle	Flux (L/m^2^·h)	Mean Roughness (nm)	%Rejection	Ref.
PSf	ZnO-GO 0.6%	90%	4.09	39.6°		4.03	99% humic acid	[63]
PES	0.1 wt% ZnO	47.34%	13.96	60.9°	80 kg/m^2^·h	24.74	94% humic acid	[20]
PVC	3 wt% ZnO	79.8%	12.1	54.5°	401.9 kg/m^2^·h		97.5% BSA	[74]
PPSU	(0.025) ZnO/MWCNTs		120.43	57.5°		2	98.57% Direct red 80	[75]
PES	ZrO_2_ (1% wt)			52.3°	83.6 L/m^2^·h	13.8	92.7% BSA 91.2% OVA	[70]
PES	TiO_2_/F127	91.3%	22.09	61.2°	235.9 L/m^2^·h	4.24	96% BSA	[92]
PES	CuO/ZnO (0.2%)		39.76	65.5°	679 kg/m^2^·h	8.19	99% BSA	[18]
PES	CuO (0.1%wt)		16.7	64°	869.9 kg/m^2^·h	1.34	97% BSA	[23]
PPSU	2 wt% BiOCl-AC	72.99%		67.40°	465.35 L/m^2^·h		80% diesel fuel & 90.74% crude oil	[93]
PVC	TiO_2_(1.5 gm)	79.5%	77	62.5°	116 L/m^2^·h		96.3% oil and grease 79.7% COD 98.8% TSS	[11]
PVDF	TiO_2_ (<2 wt.%)		47.3	76°	111.7 L/m^2^·h		100% BSA	[94]
EPVC/PEG	TiO_2_ (2 wt.%)	78.7%	25	57.2°	435 kg/m^2^·h		98% BSA	[95]
PES/PVP	Ni@MXene (1 gm)			54.15°	1181 L·m^−2^·h^−1^·bar^−1^	24.3	64.6% BSA solution 99.8% HA solution	[61]
PSF/PVP	MXene nanosheets (500 mg/L)	74.4%	36	78.4°	306 L·m^−2^·h^−1^	15.5	98% BSA rejection	[82]
PES	MOFs@GO (UIO-66@GO) 3.0 wt.%			60.4°	15.5~kg/m^2^·h	11.8	98.3% dye rejection	[88]
PSF	Tp-TTA/mPSFx COFs 10%		1.6	42°~	36.52 L m^−2^ h^−1^ bar^−1^		98.18% CB-T rejection	[91]
14%PES/2%PVP	ATO 4 wt.%		9.2	47°	22–15 L·m^−2^·h^−1^		75% separation of organic matter from the BFW	[76]
PES/PVP	NaX-Zeolite(0.9 wt.%)	45.5%		63°	88.05 L·m^−2^·h^−1^		97% Pb(II) removal	[25]
PES/PVP	NaY zeolite(0.15 wt.%)	73.6%		27.68	97.8 L·m−2·h−1		99.2% removing 137 Cs ions from a liquid radioactive sample	[67]
PES	5% PTGM	81.21%		54.91	203.1 L·m^−2^·h^−1^		93.8% BSA 95.6% SA	[96]
PES/PEI	(0.7 wt.%) SiO_2_			19.11	39.7 L·m^−2^·h^−1^ 43.3 L·m^−2^·h^−1^	14.7	94.1% Congo red dye 90.5% reactive black dye	[66]
PPSU	0.15%Gum Arabic-Graphen			95.57	82.11 L·m^−2^·h^−1^		88% sodium alginate	[97]

#### 2.1.6. Carbon-Based Nanomaterials

##### GO

GO is a two-dimensional nanomaterial made of the chemical oxidation of natural graphite to different levels using the Brodie, Staudenmaier or Hummers methods or any modified method [98,99,100,101]. Figure 2, shows the structure of GO with its functional groups. The Hummers method includes treating graphite with potassium permanganate (KMnO_4_) and sulfuric acid (H_2_SO_4_), while Brodie’s and Staudenmaier’s methods employ a mixture of potassium chlorate (KClO_3_) and nitric acid (HNO_3_) to oxidize graphite [102]. On the other hand, GO can be dispersed in an aqueous media due to its hydrophilic character, and it is difficult to extract from the solution using standard separation processes. The formation of GO is affected by many factors, for example, adding a strong oxidant to the synthesis process may contaminate the graphene, which makes it useless for practical uses. Additionally, the deep oxidation of the raw graphite breaks the inherent perfect structure of graphite, resulting in unrecoverable defects in graphene structure and significantly reducing its electron conductivity [103]. Using GO in adsorption may increase the cost of industrial applications and also pollute the treated water because it is difficult to separate [98].

The inclusion of GO sheets into polymers appears to make it highly impressive for a variety of applications because of its extraordinary properties, including two-dimensional structure, ability to generate negative surface charges, excellent electron transport, high surface area and excellent chemical stabilities. Adding GO to the casting solution of membranes affects the membrane roughness, surface hydrophilicity, separation performance, permeation flux, mechanical strength and membrane fouling resistance [30]. GO derivatives are the most appealing materials for membrane separation procedures because of their great accessibility, chemical stability and mechanical strength. GO derivatives can be produced using oxidation, exfoliation and reduction of GO. GO and GO derivatives have attracted interest as nanofillers in membrane applications because of the existence of oxygen functional groups, such as hydroxyl, carboxyl, carbonyl and epoxy groups. Functionalized GOs, such as GO nanoplate, rGO and sulfonated graphene oxide (sGO), have a unique physical morphology, which makes them contribute to different membrane geometries and separation performance and features [104].

In recent years, using GO nanofillers in the synthesis of polymeric membranes has highly increased due to their impact on the membrane’s development. However, these membranes are not ready for industrial production due to a number of difficulties that have not yet been resolved. Figure 3, shows that the number of publications that utilize GO in membrane modifications increased dramatically in 2011–2020. Since, 765 results were found using the keywords “graphene oxide,” “membrane” and “water separation,” with China leading the way, followed by the US and India [104].

The GO NPs’ effectively modify membrane pores during membrane synthesis by the phase inversion process. In comparison with the structure of the neat PES membrane, all the hybrid membranes have larger pore shapes. According to Ng et al. [104], the structure of the top layer of a membrane that did not contain GO NPs was dense. The pore channels of the MMMs became larger and wider after being embedded with GO NPs (0.5 wt.% GO), creating a tortuous structure in a PES membrane. Mahalingam et al. [105] reported the possibility of obtaining an extremely thin skin layer even after adding GO NPs as it changes the surface morphology of the membranes free of small drawbacks. The obvious change that occurred after adding GO is changing the porosity and the membrane’s color from white to dark brown, confirming the uniform distribution of GO throughout the polymer matrix. The performance of membranes with GO nanofillers has been studied using a number of GO-related investigations. The incorporation of GO into a hydrophobic polymer membrane could improve the membrane’s wettability by increasing the surface hydrophilicity and thus improve the permeability and antifouling properties. Previous studies have shown that adding GO to a membrane’s casting solution improves the antibacterial properties, water flow, fouling resistance and solute rejection [99,104].


**Pure GO additives**


Table 3 shows that the inclusion of GO NPs successfully changed the properties of PES and PVP UF membranes. Introducing GO NPs into the PES casting solution affects the membrane’s morphology and considerably increased the mean pore radius. The pure water permeability (PWP) of the membranes increased when the GO NPs were added because of the hydrophilicity of the added NPs since all the prepared membranes showed high dye removal above 99% at dye concentrations of 10, 50, 80 and 100 ppm and operating pressure of 3 bar [30]. According to Table 3, PVC/GO membranes were made by the phase inversion technique for oily wastewater treatment. GO was employed to solve the fouling problems with PVC membranes because of GOs’ high surface characteristics [106]. PVDF/GO membranes with no PVP appeared to have a more compact skin layer than the pure PVDF membranes. With increasing PVP concentration, the length of the finger-like porous sub-layer expanded.

The pores of the PVDF/GO membrane are larger than those of the pure PVDF membrane pores. The membrane surface shape clearly changed after the inclusion of GO into PVDF membranes, and the convex sections became significantly smoother. Therefore, the hydrophilicity and the anti-fouling performance have clearly been enhanced by the incorporation of GO into membranes [107]. The incorporation of GO in PPSU enhances the membrane’s pore structure. When the GO concentration was 1.5 wt.%, the pure water flux of the prepared MMM reached 231.7 L/m^2^·h, with an increase of 83% over the pure PPSU membrane. However, the BAS removal was not high. Water molecules would prefer to interact with the polar groups on the membrane’s surface containing GO, which hindered the pollutant and boosted the membrane’s anti-fouling resistance [108].

Adding GO to the casting solution of PSF UF membranes increased the wetting, rejection and antifouling properties [109]. The produced membranes had higher hydrophilicity and porosity, which improved the permeability, flux and lead ion rejection when compared to the GO-free membrane. The maximum rejection was 98% at 1 bar and it declined when the GO concentration decreased. The studies showed the importance of GO in generating highly porous membranes that provide a better flux and lead ion rejection without compromising membrane performance [110].


**Hyper-GO into polymeric membrane**


Several research papers have concentrated on GO NPs in recent years [30]. The main purpose of this combination is to combine the good characteristics of two types of materials, hence achieving better effectiveness. Mixed NP research had attracted more attention in the last decade than mixed polymers because they offer an outstanding performance in eliminating specific impurities and fouling problems [111]. Embedding nanofiller into a porous polymeric substrate or thin-film membrane could improve foulant resistance and selective separation and reduce membrane fouling [104]. Table 4 presents some of the most relevant published works. The inclusion of oxygen-functional hydrophilic groups on the surface of GO, such as hydroxyl, carbonyl and carboxyl groups, can increase the hydrophilicity of GO and improve the properties of GO/polymer hybrid materials. Many functional nanohybrids have already been created by adding numerous NPs onto the surface of GO, including SiO_2_ [112], Au [113], Ag [114], ZnO [29] and TiO_2_ [115,116,117]. A combination of the above-mentioned components could exhibit unusual antibacterial characteristics to inhibit the bio-fouling effect. The incorporation of Ag–GO nanocomposites into the PES membranes improved the hydrophilicity of a membrane surface, pure water flux and rejection [104].

PSf hybrid membranes were fabricated by the inclusion of hydrophilic SiO_2_–GO in the casting solution. The SiO_2_ was uniformly distributed over the GO surface to form the SiO_2_–GO nanohybrid before being added to the PSf matrix to obtain a compatibility-enhanced inorganic polymer. The water flux of the membranes increased dramatically with increasing the amount of SiO_2_–GO added to the membrane. At 0.3 wt.% SiO_2_–GO, the permeate reached nearly twice that of the PSf membrane, although the rejection of egg albumin remained high at >98% [112]. A study conducted by Sadiq et al. [48] reported the considerable influence of MWCNT-g-GO on the surface morphology of the membranes. The contact angle, roughness and porosity of the membranes were enhanced by adding 0.119 wt.% of MWCNT-g-GO to the casting solution.

The fouling problem of PVC membrane was solved by adding combinations of both GO and ZnO NPs to the PVC matrix. Since the presence of ZnO NPs eliminates the problem of aggregation of GO in a PVC matrix, thus obtaining a favorable connectivity between finger-like pores and macro gaps. Additionally, the excellent antifouling properties that both GO and ZnO NPs have and add to the MMMs are reflected in the remarkable turbidity removal and permeate flux [106]. ND NPs agglomerate when employed as fillers in MMMs; therefore, GO nanoplatelets were used to eliminate this problem by uniform dispersion of ND NPs on the high-surface-area GO nanoplatelets. The inclusion of a combination of ND–GO in the PVC matrix formed MMM with high hydrophilic properties led to improve water flow, rejection and mechanical properties [118]. Same improved properties were obtained for a PVDF/TiO_2_–GO membrane used for BSA removal [119].

Chung et al. [63] reported the preparation of PSF MMMs using nanosheets made of ZnO–GO to obtain a hydrophilic membrane based on the obvious polar properties of ZnO and the presence of hydroxyl, carbonyl and epoxy groups in GO structure. The prepared MMMs performed well even after reducing the amount of ZnO NPs dispersed on GO nanosheets to five times. In comparison to other GO NP composites with TiO_2_, SiO_2_ and Ag, the GO–ZnO/PSF composite membranes seem to be the most hydrophilic and have the lowest contact angle value. Additionally, the membrane demonstrated substantial improvement in terms of permeability, porosity, pore size, rejection tendency and fouling propensity. In addition, the prepared MMMs were tested using E. coli for bio-fouling problems. The prepared GO–ZnO/PSF composite membranes were superior membranes with high hydrophilicity and fouling control, and they are ideal for a variety of separation and purification applications.

The addition of hydrophilic filling GO NPs and amphiphilic copolymer PF127 (macromolecular polymer additive) and PEI matrix improved surface hydrophilicity, decreased the oil fouling and enhanced the membrane performance in oil–water separation. This composite membrane demonstrated a high increase in the surface porosity, which enhanced the permeate flux. This composite membrane retained over 95% of oil in the oil–water emulsion and showed high recyclability and antifouling [120]. Liu et al. [121] studied the separation of a dye using the unique prepared hydrophilic PDA/RGO/HKUST-1 membrane with good antifouling properties. The inclusion of the (HKUST-1) into GO nanosheets increased interlayer spacing, which improved the separation efficiency and flux of the PDA membranes.

Figure 4 shows the total number of publications for using GO in UF membranes’ all over the world in 2010–2021 in Elsevier. During these twelve years, the number of publications increased, and the largest number of published articles was in 2021, which was 8622 research articles for GO. Figure 5 shows the number of publications using GO in the field of UF membranes in Google Scholar, utilizing GO for UF membranes increased during the last twelve years to reach 4520 in 2021. These numbers were obtained based on a search conducted in ELSEVIER.

Although GO NPs enhance the fouling resistance, they have poor antibacterial characteristics, which could lead to bio-fouling on the membrane surface, especially when used for wastewater treatment where microbes are abundant. Thus, metal oxides or inorganic particles with antibacterial properties could be incorporated into GO NPs to reduce the likelihood of membrane bio-fouling.

##### Carbon Nanotubes (CNTs) and Nanofibers (CNFs)

Every carbon-based nanoparticles has benefits and drawbacks, and the recommendation of one material over the other is primarily based on improved properties, and environmental and health concerns. CNTs are a type of carbon network with a one-dimensional cylindrical nanostructure and high thermal conductivity, tensile strength as well as electrical properties. CNTs’ high-specific surface area and oleophilic properties offer important advantages for the development of “oil-removing” membranes with large permeation flux [122]. Wang et al. developed f-CNT/PES ultrafiltration membranes, with CNTs being functionalized by sodium lignosulfonate (SLS). Wang et al. found the composite membranes’ finger-like structure, and the placement of f-CNTs on the membrane resulted in enhanced membrane hydrophilic nature with reduced surface roughness. Water flux as well as fouling protection of the f-CNT/PES composite membranes were significantly improved. Through three stages of antifouling experiments, the f-CNT/PES composite membranes had the highest FRR and Rr. Additionally, the BSA removal rates for all manufactured membranes were greater than 95%. Antibacterial characteristics are not present in any of the produced membranes. However, when a modest electric field was applied, the CNT/PES as well as f- CNT/PES membranes demonstrated good synergistic antibacterial activities [123].

CNFs and CNTs have identical electrical and mechanical properties; however, CNFs have a significantly greater functionalized surface area than CNTs. Electrospinning can be used to manufacture CNFs, which have larger porosity, constant pass-through size and an advantage of this system porous structure [122]. Utilizing electro-spinning technology, Liu et al. produced a sublimation technique for producing macro-porous CNFF (MCNFF) employing PTA as a sublimated agent. PTA sublimed and formed macro-pores inside the carbon nanofibers during the post-thermal treatment of PTA-PAN composite nanofibers. Liu et al. found that the MCNFF has an increased in porosity, flexibility, as well as high stability and achieved selective oil absorption from water with high uptake capability [124].

## 3. Effect of NPs on the Morphology of the PPSU, PES and PVC Membranes

Figure 6 shows the SEM characteristics of PPSU membrane with and without the inclusion of NPs. In Figure 6A, for the pristine PPSU, fewer pores were observed at the membrane surface, while in Figure 6B, with the addition of 1.5 wt.% GO, more pores with more uniform size were formed on the surface, and the membrane surface became very smooth; therefore, both are favorable to enhance the membrane flux [108]. The addition of 0.025 wt.% ZnO NPs to the casting solution modified the PPSU membranes’ structure to a totally sponge-like structure, as shown in Figure 6C. Because of the presence of ZnO NPs in the polymer solution, this behavior is related to a delayed liquid–liquid demixing process between the polymer solution and the non-solvent (water). Additionally, it seems to have a larger pore size and rougher surface [73]. In Figure 6D, at 20 µm magnification, the spherical shape of TiO_2_ NPs with anatase phase in the form of dots can be seen to be randomly scattered throughout the membrane surface with no aggregation. The cross-sectional morphologies indicate the finger-like macrovoid pore structure, which might be attributed to the hydrophilic nature of TiO_2_ interacting with water molecules during phase inversion [125].

In Figure 7B, it can be observed that the pores for the membranes embedded with GO are rather larger than those of the neat PES membrane, as shown in Figure 7A. The hydrophilic property of GO enhances the mass transfer rate between the solvent and the non-solvent during phase inversion and led to the creation of bigger pore channels [126]. In Figure 7C, the morphologies of cross sections were unaffected by ZnO NPs. The surface morphologies of the resulting membranes altered from a convoluted surface with straight circular holes to a comparatively flat surface with straight circular pores. Furthermore, a progressive rise in pore size and density was observed [127]. When TiO_2_ is added to the polymeric solution, the macrovoid dimensions increase, while the sponge-like structure of the membrane is suppressed. The cross-section image in Figure 7D represents the TiO_2_ concentration effect, and two different effects can be seen: As the length of the macrovoids increases to roughly 20 µm, a construction with fewer macrovoids is formed [128].

The neat PVC in Figure 8A has a typically asymmetric, extremely porous and non-homogenous shape with a thick top layer that is responsible for penetration and rejection. The surface of the PVC membranes that have been treated with GO in Figure 8B becomes rougher than the surface of pure PVC membranes. The pore-like structure on the surface of modified PVC membranes can clearly be seen in the SEM image. The quantity of holes in the composite membranes increases with ZnO addition, while porosity is currently higher with ZnO concentration in ZnO/PVC membranes as could be seen in Figure 8C. Furthermore, the connectivity between the top and bottom layers improves with the inclusion of ZnO. It seems, in Figure 8D, that adding the TiO_2_ concentration in the membranes increased both the number and total area of pores.

Each nanoparticle was added for a different reason, either to add a specific feature or to improve one of the properties of the membrane. It could be seen that every additive on a polymeric membrane makes a difference in the morphology of the surface and pores. This change could help in enhancing the properties of the membrane separation by creating more larger pores on the surface, making a smooth surface and improving the hydrophilicity of the polymeric membrane, to increase the flux and rejection of the membrane. It must be considered that these additives should be added according to the required dose because they will agglomerate on the surface and block the pores; so, they can make the surface thicker, which decreases the flux and the rejection of the membrane and these additives can then become a defect instead of improving performance.

## 4. Application of MMMs

In recent years, membranes have been considered a new technology having a several applications in many sectors as shown in Figure 9, not only for wastewater filtration but also for gas separation, such as CO_2_ extraction from gas mixtures, removal of dyes [130,131] and for gas storage in biogas plants.

Polymeric membranes are considered the cheapest technology and quickly available. There are many research papers that use polymeric membranes in the environment because it uses less energy than other separation technologies and even operates without heating. So, many researchers tried to modify the polymeric membrane by adding different additives to make a special membrane for each application. One of these additives is the NPs that have been studied in recent years by adding them to the polymeric membrane. Each NPs have special properties that could be used for special applications. For example, ZnO, Ag, TiO_2_ and CuO are considered as antimicrobial NPs that can be added to different polymers to be used in medical applications because of their benefits in the treatment of bacterial infections. For biomedical applications, iron oxides particles, such as magnetite (Fe_3_O_4_) or its oxidized counterpart maghemite (Fe_2_O_3_), are most typically used.

Zhao et al. achieved a greater flux recovery with increasing GO content due to the higher selectivity of the PVC/GO hybrid membranes for water molecules. For example, the permeability recovery ratio of the composite membranes blended with 0.10 and 0.15 wt.% GO was 70.4% and 75.9%, respectively, which is much greater than even the pure PVC membrane (only 41.8%), indicating a superior antifouling ability [129], as shown in Figure 10A.

Igbinigun et al. showed that the commercial PES-UF membrane had the greatest flux decreases, most likely as a result of increased fouling generated by interactions between the hydrophobic regions of the humic acid and the hydrophobic surfaces of the commercial PES UF membrane surface. However, when introducing GO, the hydrophilic and relatively smooth surface of GO membranes help to reduce humic acid foulant attachment to the GO-enhanced membrane surfaces [132], as shown in Figure 10B. Wasim et al. developed a simple method for synthesizing silane functionalized rGO embedded in cross-linked PVA for azo dye removal. After VTES-G addition, the dyes were successfully rejected in 97.8% (Congo Red), 99.9% (Reactive Black 5) and 96.8% (Reactive Orange 16), as shown in Figure 10C1–C3), respectively. In comparison to rGO, VTES-G has rough surfaces because silica spheres cover the top layer of rGO sheets, preventing aggregation and allowing for a well dispersed filler. Additionally, the decrease in oxygen-containing groups makes rGO more hydrophobic in nature, which increases its attraction for aromatic compounds via stacking. This phenomenon enables the use of decreased GO in dye removal and adsorption applications.

**Figure 10 membranes-12-01043-f010:**
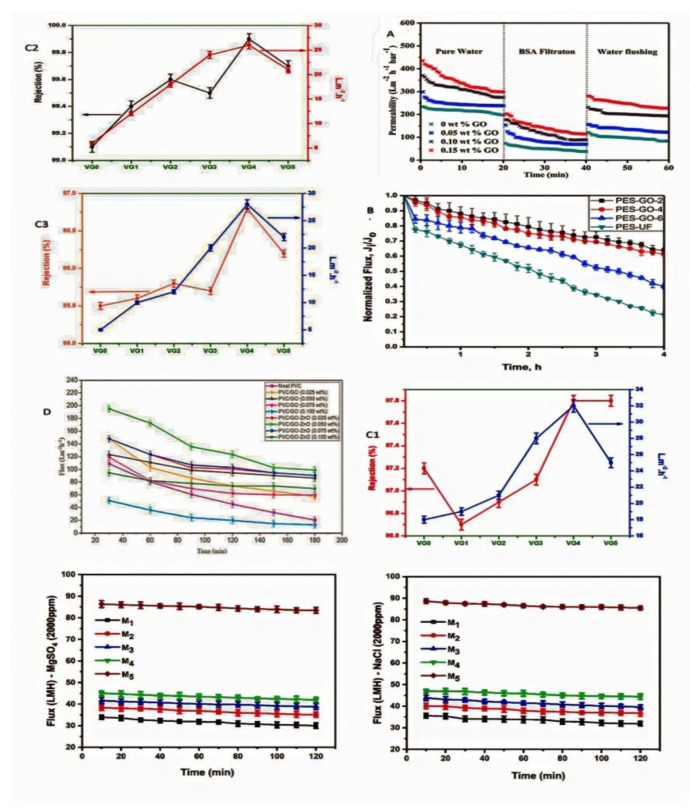
Perm−selectivity of different polymers mixed with GO. (**A**) PVC/GO membranes filtration of BSA solution [129]. (**B**) PES/GO membranes humic acid filtration [98]. (**C**) PVA/VTES−G membrane for dyes: (**C1**) Congo Red, (**C2**) Reactive Black 5 and (**C3**) Reactive Orange 16 [133]. (**D**) PVC/GO−ZnO membrane filtration of oily wastewater [106]. (**E**) PSf/GO−vanillin membranes filtration (NaCl) and (MgSO_4_) [109].

The VTES-G was synthesized effectively using a modified Stober technique using vinyltriethoxysilane (VTES) as a precursor. The VTES performed as a silica source, forming SiO_2_ NPs during the hydrolysis and reduction process [133]. Kazemi et al. manufactured PVC/GO and PVC/GO-ZnO membranes for oily wastewater treatment by using the phase inversion technique [106]. Because of its excellent surface features, GO was developed for solving the fouling issue of PVC membrane. ZnO NPs were employed to tackle the GO aggregation problem in a PVC matrix. It showed where both nanocomposite membranes exhibit excellent antifouling efficacy, great turbidity reduction and greater flux recovery. The flux breakdown for neat PVC membrane was around 81%, as well as for membranes of 0.025, 0.05, 0.075 and 0.1 wt% GO being around 61%, 30%, 49%, and 74%, respectively, and also for membranes with 0.025, 0.05, 0.075, and 0.1 wt% GO-ZnO being around 38%, 49%, 38%, and 26%, respectively. It also indicated fouling resistance, which showed that membrane hydrophilicity seemed to have a positive effect on oily wastewater filtration, and that increasing the hydrophilicity improved the antifouling capabilities of the membranes.

Yavad et al. passed 2000 ppm NaCl as well as MgSO_4_ aqueous solutions through the membrane at constant 5 bar pressure for salt rejection studies. According to the observed measurements, the permeate flux improved 2.5 times from 35 to 88 LMH for M1 and M5 membranes. The decrease in permeation flux throughout the operation might be related to surface/pore adsorption caused by electrostatic contact between the membrane and the foulants. For MgSO4 and NaCl, the M4 membrane had the largest rejection of 92.51 ± 2.73 and 25.43 ± 3.12, respectively. M3 and M4 (PSf/GO–vanillin) membranes rejected divalent ions 4% more than M1 (PSf-vanillin) membranes, maybe due to the increased negative surface charge for M3 and M4 membranes attributed to the presence of GO molecules. The inclusion of GO enhances the hydrophilicity of a membrane, resulting in less fouling matter contact with the membrane surface [109].

The extraction of heavy metals, such as Hg, Pb, thallium, cadmium and arsenic, from natural water has received considerable interest due to their negative impact on the environment and human health. NPs such as iron oxide NPs can be used as additives for the polymeric membrane to be an effective sorbent material for toxic materials. Researchers are showing great interest in metal oxides and ceramic NPs due to their applications in fields, such as catalysis, photocatalysis [134,135], photodegradation of dyes and imaging applications. It is worth mentioning that fuel cell membranes have received a lot of attention in recent years, especially membrane fuel cells that use a solid polymer as an electrolyte. This technology has the greatest potential for energy conversion in electric equipment and portable or fixed systems, which can be modified with platinum, ruthenium, Ag and ZrO_2_ NPs, and carbon NPs are also employed in the construction of fuel cells. All these NPs are added to the polymeric membrane to increase the hydrophilicity of the membrane so as to increase their flux and the rejection of the membrane. Additionally, improving chemical and physical properties can be appropriate for specific applications.

Due to the enhanced permselectivity, higher hydrophilicity, and enhanced fouling resistance, mixed membranes with nano-sized materials, e.g., MWCNT, Ag CuO and GOs, have recently attracted the interest of researchers in the water treatment field. TiO_2_ is used in the photocatalytic degradation of organic pollutants in waters because of their photocatalytic activity in the solar spectrum, high stability and selectivity, and low cost. Before discharging wastewater into any stream, it must be treated. Although the photodegradation of these organic compounds in the presence of a photocatalyst is a well-established approach, developing effective, affordable and reusable catalysts remains difficult [135]. They are also considered as antibacterial NPs and efficient antimicrobial coating due to TiO_2_′s ability to transfer charge and photoinduced oxidation. L. Zhang discovered that the TiNA200-PTFE composite membrane kills bacteria primarily by the physical piercing of cells by sharp crystallization vertices, followed by oxidative from the sun-mediated formation of hydroxyl radicals while membrane cleaning (Figure 11) [136].

Ag and CuO are considered as antimicrobial NPs that have been added to different polymers for use in the medical section because of their benefits in the treatment of bacterial infections. As shown in Figure 12, blended membranes (1 wt.% Ag-Cu_2_O NWs/PSF) were evaluated for antibacterial activity against Gram-negative Escherichia coli (abbreviated as E. Coli) and Gram-positive Staphylococcus aureus (abbreviated as S. Aureus). Xu et al. showed that pristine PSF surfaces were incapable of killing microorganisms. The Cu_2_O mixed membranes were quite low in toxicity, with only faint zones surrounding the discs and minimal formation of microbial colonies, whereas for Ag–Cu_2_O NWs/PSF (Figure 12a,b), as well as Ag NPs/PSF (Figure 12e,f), the blended membranes showed outstanding antibacterial action against both Gram-negative E. Coli and Gram-positive S. Aureus. The clear zones surrounding the discs were not seen. Studies have shown that the amount of Ag in the membrane plays an important role in its antibacterial properties [137].

## 5. Conclusions and Future Prospects

There is a need to perform the removal of various pollutants from different wastewaters to be safely discharged into the environment. This review focused on the importance of the effects of various NPs embedded into different polymeric membranes on the removal of pollutants from wastewater. Most recent research on MMMs has concentrated on how to develop the membrane structure and surface properties by NPs and thus on membrane performance. The functions of NPs and their impacts when embedded with the membrane, such as hydrophilicity, self-cleaning and antifouling to increase permeability, have been evaluated. Common NPs embedded into the membranes include GO, hyper GO, TiO_2_, and some of the recent nanoparticles such as MXene, MOFs, and some metal oxide NPs. It can be concluded that NPs could be an excellent choice for improving the treatment of various wastewaters.

For the application for each membrane composite with different NPs, we can conclude the following:The addition of GO may make the membrane more adsorbent and have good anti-fouling properties by increasing the membrane hydrophilic character. This is attributed to GOs’ characteristics, which include a large surface area and assessable adsorption sites, higher hydrophilic properties and selective site.ZnO is used in photocatalytic membranes for organic pollutants in water because of its photocatalytic activity, stability, selectivity and low cost. It could also be used with membranes in medical application due to the antimicrobial properties that have been added to various polymers since it has been confirmed to be useful in the treatment of bacterial infections. The same is true of Ag, TiO_2_ and CuO, which are also used with polymeric membrane in the medical sector.Ag, Ni and Zr are employed in membranes for the separation of gases. For example, ceramic dense membranes are used in the separation of oxygen from air or even the separation of hydrogen from a mixture of gases. The low permeability limits their industrial applications.WOx mixed with membranes has been widely used, especially in recent years, because of its special properties: photocatalytic application and antimicrobial use. This has become one of the most studied metal oxides.Because of the efficiencies seen and the advantages of MOFs, COFs, MXene nanoparticles, researchers have paid considerable attention to mixing them with polymeric membranes and this research has grown rapidly in recent years.MXene/polymer membranes have been presented and shown great promise, with an overall performance superior to neat polymer films.COF materials with ordered channels and functionalized groups inside the channels provide a new strategy for achieving high performance in the advancement of membrane processes for separation.MOFs, compared to the ordinary inorganic particles with ‘rigid’ frameworks, have a unique nature that may support growing with the polymer, allowing for good compatibility.

This study showed that nanoparticle additives can be used as single additives or composites to provide the membrane different properties that make it suitable for a particular application. The second main conclusion of this review is the problems that occur to the membrane and its properties when the hydrophilic polymer membrane is converted to a super hydrophilic polymer membrane. Moreover, in the literature, there is a lack of experimental data in industrial wastewater treatment using NPs, and most experimental works are performed on a laboratory scale. Accordingly, more experimental work should be performed in this field to evaluate the effect of NPs in MMMs on the removal efficiency of pollutants.

## Figures and Tables

**Figure 1 membranes-12-01043-f001:**
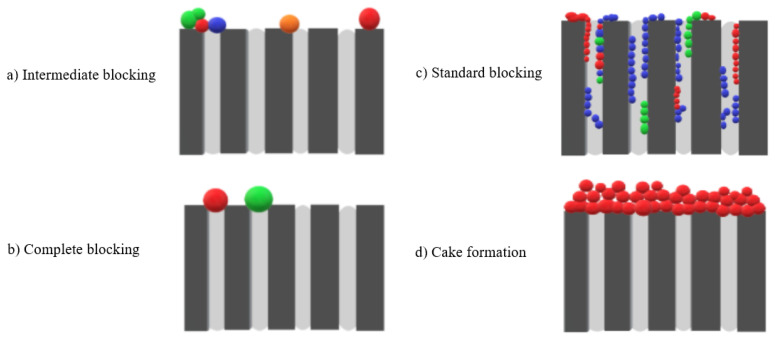
Fouling mechanisms (**a**) Intermediate blocking, (**b**) Complete blocking, (**c**) Standard blocking, (**d**) Cake formation.

**Figure 2 membranes-12-01043-f002:**
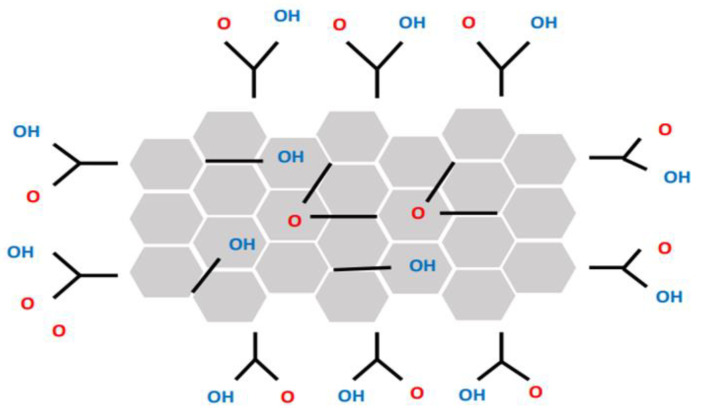
The structure of graphene oxide (GO).

**Figure 3 membranes-12-01043-f003:**
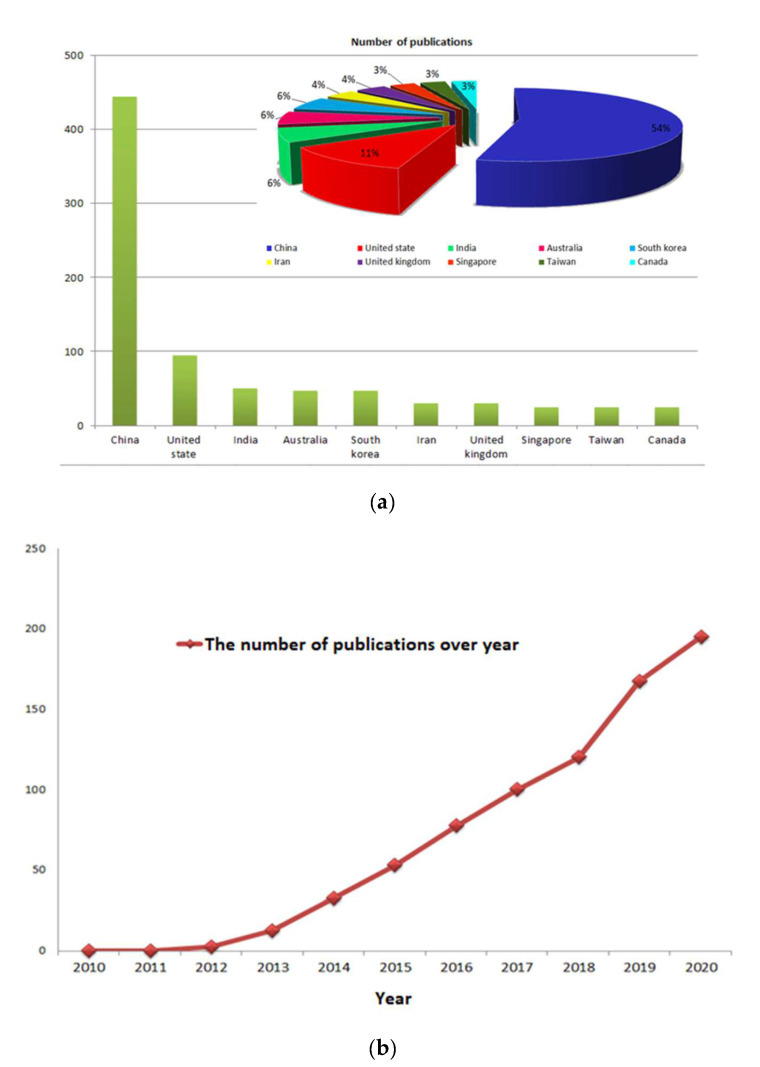
The number of publications that are related to this research topic. Data were obtained from the Scopus database from the years 2011–2020 [104]. (**a**) Percentage of publication according to the country of the authors; (**b**) Publications over year.

**Figure 4 membranes-12-01043-f004:**
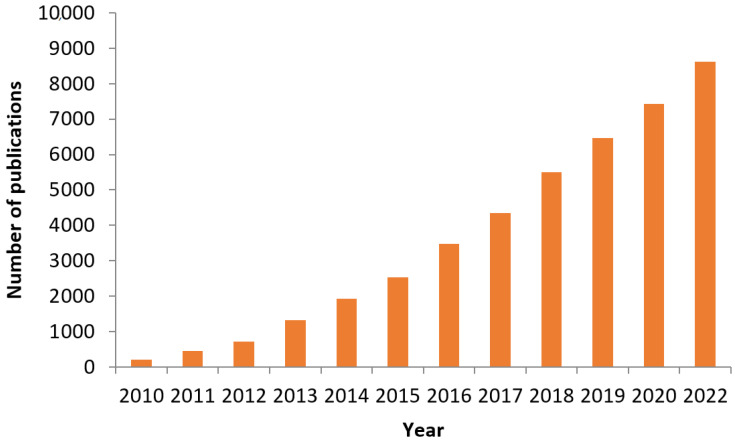
The total numbers of publications using GO in 2010–2021 (ELSEVIER).

**Figure 5 membranes-12-01043-f005:**
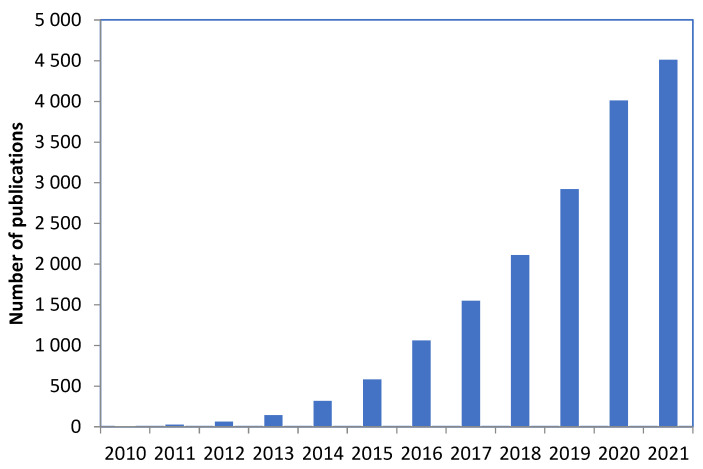
The number of publications of GO in UF membranes in 2010–2021 (Google Scholar).

**Figure 6 membranes-12-01043-f006:**
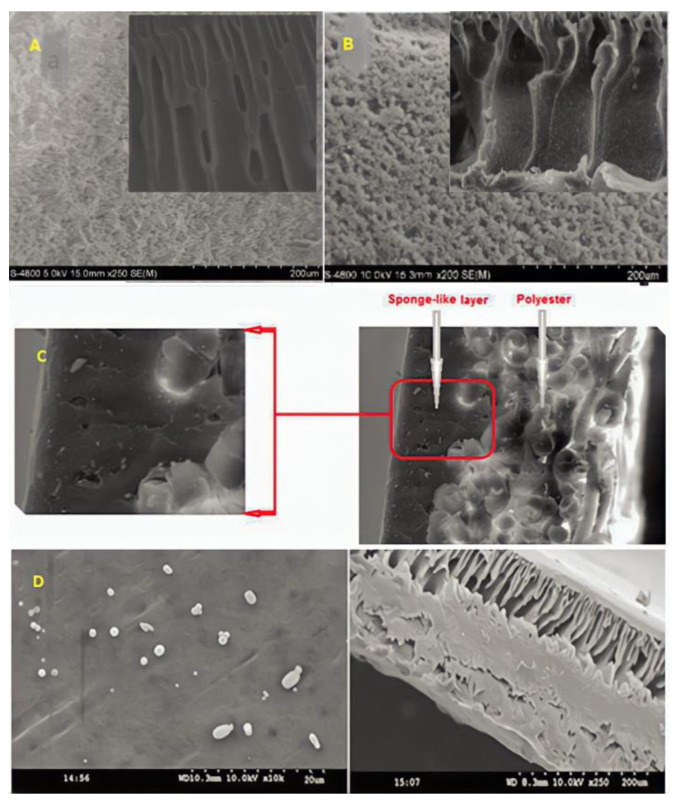
SEM images of the PPSU membrane with different addition of nanoparticles. (**A**) Pure PPSU [108], (**B**) 1.5 wt% GO [108], (**C**) 0.025 wt% ZnO [75], and (**D**) 0.54 wt% TiO_2_ [125].

**Figure 7 membranes-12-01043-f007:**
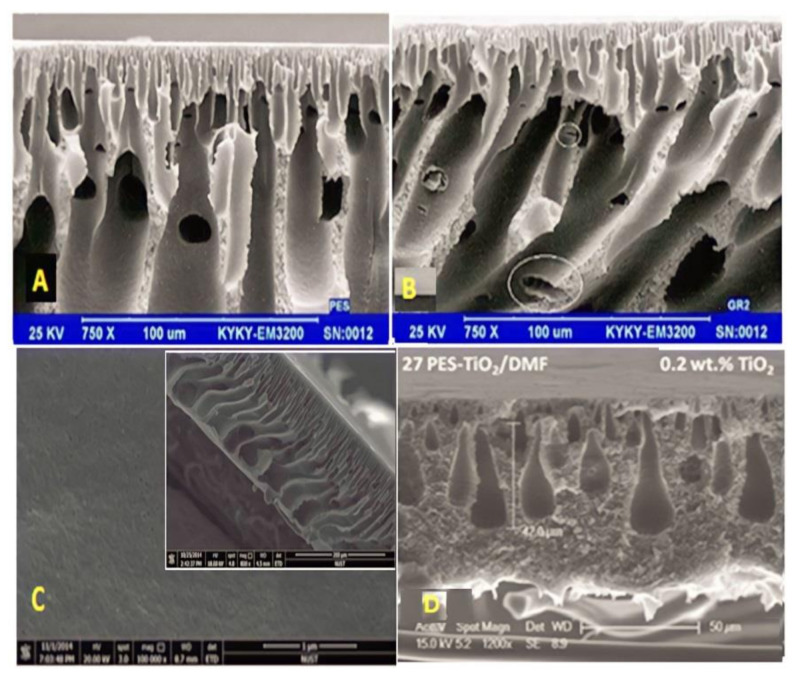
SEM images of the PES membrane with different addition of nanoparticles. (**A**) Pure PES [126], (**B**) 0.5 wt% GO [126], (**C**) 0.75 wt% ZNO [127], and (**D**) 0.2 wt%TiO_2_ [128].

**Figure 8 membranes-12-01043-f008:**
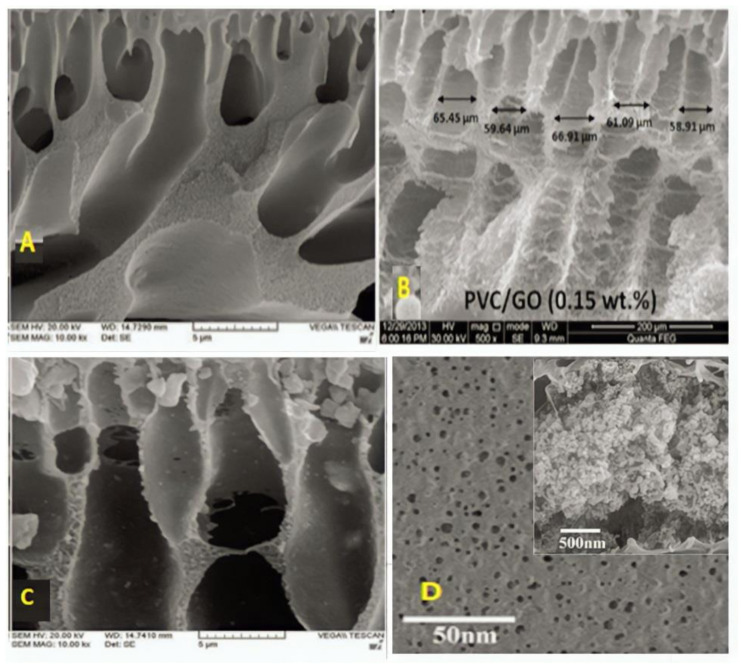
SEM images of the PVC membrane with different addition of nanoparticles. (**A**) Pure PVC [74], (**B**) 0.15 wt% GO [129], (**C**) 3 wt% ZnO [74], and (**D**) 2 wt%TiO_2_ [117].

**Figure 9 membranes-12-01043-f009:**
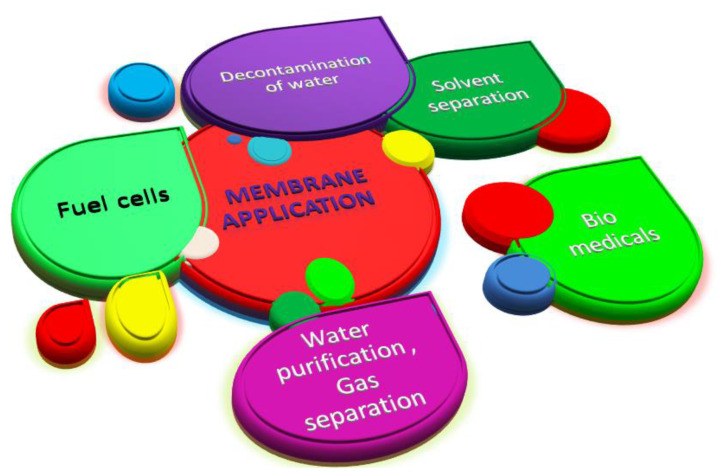
Membranes’ environmental applications.

**Figure 11 membranes-12-01043-f011:**
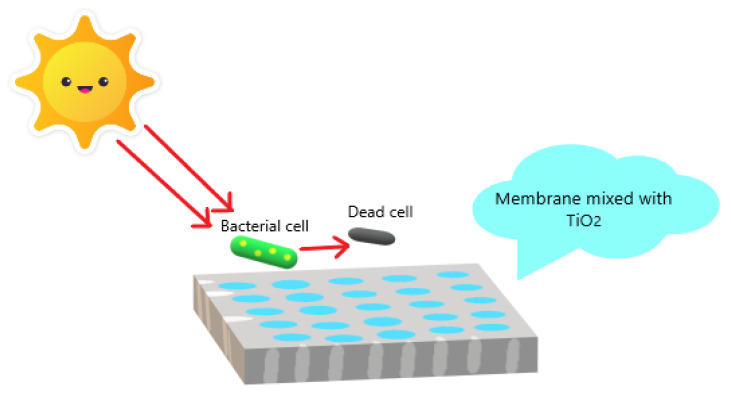
Photocatalytic membrane (polymeric membrane with TiO_2_).

**Figure 12 membranes-12-01043-f012:**
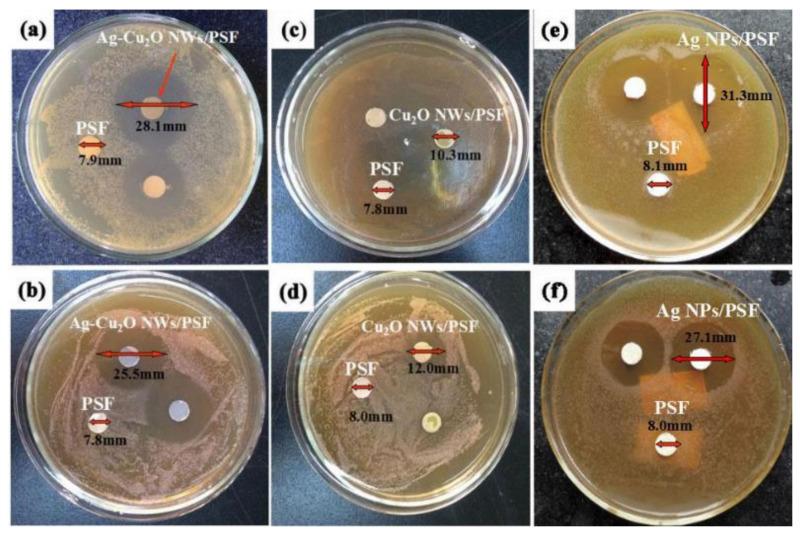
Inhibition zone of membranes with (**a**,**c**,**e**) E. coli and (**b**,**d**,**f**) S. Aureus [137].

**Table 1 membranes-12-01043-t001:** Hermia’s model (fouling mechanisms).

Fouling Mechanisms	V-t Equation	J-t Equation
Complete pore blocking	$V=\frac{-Q_{0}}{K_{b}}\left(1-e^{-K_{b}t}\right)$	$\ln\left(J\right)=\ln\left(J_{0}\right)-K_{c}t$
Standard pore blocking	$\frac{t}{V}=\frac{K_{s}}{2}t+\frac{1}{Q_{0}}$	$\frac{1}{J^{1/2}}=\frac{1}{J_{0}{}^{1/2}}+K_{s}t$
Intermediate pore blocking	$V=\frac{1}{K_{i}}\ln\left(1+K_{i}Q_{0}t\right)$	$\frac{1}{J}=\frac{1}{J_{0}}+K_{i}At$
Cake filtration	$\frac{t}{V}=\frac{K_{C}}{2}V+\frac{1}{Q_{0}}$	$\frac{1}{J_{2}}=\frac{1}{J_{0}^{2}}+K_{C}t$

**Table 3 membranes-12-01043-t003:** Type of polymers with pure GO additives.

Type of Polymer	Composition of Pure GO	%Porosity	Pore Size (nm)	Thickness (μm)	Tensile Strength (Mpa)	Contact Angle	Flux (L/m^2^·h)	Rejection (%)	Ref.
PES	PES–PVP–0.5 GO	80.6%	14.59	150.13	2.55	39.21	116.5 LMH bar	99.7% Acid Black dye	[30]
PVC	PVC/GO (0.05)		55		2.86	75.51	1526.71 L/m^2^·h	96.62% Oil	[106]
PVDF	PVDF/GO-PVP					68	104.3 L/m^2^·h	85% BSA	[107]
PPSU	1.5 wt% GO	63.7%	30.1			67.1	231.7 L/m^2^·h	95% BAS	[108]
PSF	200 mg GO	56%	10.58	101		50.31	91 L/m^2^·h	92.5% MgSO_4_	[109]
PSF	1% wt GO	82.23%	22.73		1.1	34.2	163.71 L/m^2^·h	25.4% NaCl	[110]

**Table 4 membranes-12-01043-t004:** MMMs made of different polymers and combined GO–inorganic NPs.

Type of Polymer	Composition of GO–Inorganic NPs	Porosity (%)	Pore Size (nm)	Tensile Strength (Mpa)	Contact Angle	Flux (L/m^2·^h)	Rejection (%)	Ref.
PVC	PVC-0.119 wt.% MWCNTs-g-GO	81.4	259	1.4	13.9	254 L/m^2^·h	88.9% Oil	[48]
PVC	PVC/GO-ZnO (0.05)		25	3.49	68.78	1255.14 L/m^2^·h	99.55% Oil	[106]
PVC	PVC/GO-ND 0.1))		350	5	64.6	425 L/m^2^·h	95.08% BSA	[118]
PVDF	PVDF/TiO_2_–GO	43	50		67	199.97 L/m^2^·h	91.38% BSA	[119]
PEI	PEI/PF-127/GO (0.6 wt%)	76.8	28.23	5.9	46.8	325 L/m^2^·h	95% Oil	[120]
PDA	PDA/RGO/HKUST-1				32.1	184.7 L/m^2^·h	99.8% dye	[121]
PSF	SiO_2_-GO/PSF		2.3		63	360 L/m^2^·h	98.3% BSA	[112]

## Data Availability

Not applicable.

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
