# Peer review of "Classification of Nanomaterials and the Effect of Graphene Oxide (GO) and Recently Developed Nanoparticles on the Ultrafiltration Membrane and Their Applications: A Review"

_membranes, 2022, doi:10.3390/membranes12111043_

Round 1
Reviewer 1 Report
The topic is interesting and the collection of references and comentaries on them are up to date and relevant. Two are my concerns:
1) English is clearly too bad. Starting by the title and on along the body of the paper.
2) The material and information conveied should be reordered with sections renamed according to content etc.
I would like to find extra information on:
a) How do these membranes compare with other hydropilicity increasing treatments?
b) What about mechanical and economic feasibility of these membranes?
Author Response
RESPONSE LETTER - Reviewer #1: membranes - 1940882
Thanks to the reviewers and editorial board that our manuscript entitled – “A potential Mixed matrix ultrafiltration membranes and their impact on the environment applications: A review” was suggested to be published after proper revision. We made the requested changes and motivated our choice and answers. Thank you for your time and effort and we hope that now this work can be accepted in this highly regarded journal.
Sincerely
Prof. Dr. Qusay F. Alsalhy
Membrane Technology Research Unit
Chemical Engineering Department
University of Technology,
Alsinaa Street No. 52
Baghdad, Iraq
Email: [email protected]; [email protected]
Reviewer #1:
The topic is interesting and the collection of references and comentaries on them are up to date and relevant. Two are my concerns:
Q1: English is clearly too bad. Starting by the title and on along the body of the paper.
Answer: Thank you for the suggestion. The English language and grammar has been improved by a native speaker.
Q2: The material and information conveyed should be reordered with sections renamed according to content etc.
Answer: Thank you for this valuable remark. It has been done as can be seen in section 2.1, Page 7, Page 11, Page 14, Page 15.
I would like to find extra information on:
- a) How do these membranes compare with other hydrophilicity increasing treatments?
Answer: Thank you for this interesting and important question. A new paragraph has been added to the manuscript as follows in Page 3, lines 103-112.
Adding hydrophilic NPs to the polymeric membranes reduces their contact angle and increases their hydrophilicity and increases the separation performance of pollutants. However, adding hydrophilic polymers, such as poly(ethylene glycol) (PEG), poly(vinyl pyrrolidone) (PVP), and cellulose acetate phthalate to the polymeric membranes works as pore formers which rises the hydrophilicity of the membranes. The hydrophilic polymer additives added to the membranes can often increase the hydrophilicity at the expense of reducing the separation performance (Kumar, Rajesha, and A. F. Ismail. "Fouling control on microfiltration/ultrafiltration membranes: Effects of morphology, hydrophilicity, and charge." Journal of Applied Polymer Science 132, no. 21 (2015)).
- b) What about mechanical and economic feasibility of these membranes?
Answer: Thank you for the useful question. The effect of added NPs on the mechanical properties of the membrane have been discussed in Tables 1, 2, 3 and the sections 2.1 in pages 4-6.
Reviewer 2 Report
membranes-1940882
This review article summarized the progress of MMMs with various NPs. The authors gave a quite detailed introduction about the NPs types, the effects of NPs on MMMs performances, and relevant applications. I admit this is enduring topic in membrane community. Dozens of similar reviews have been published in the past 5 years. My first concern is that what is the novelty of this review, comparing with these already published ones. I carefully checked the introduction section, but can not found any valuable statements about the novelty of this paper.
Another concern is the NPs types mentioned in this paper. Most of them already well summarized by extensive review articles. Please simplify the 2.1, 5, and 6 sections. Because the relevant studies are not new already. I suggest to adding some discussions about the newly emerging nanomaterials, such as MOFs, COFs, and MXene. These materials have been used as fillers for MMMs, but even limited introduced in this paper. Moreover, these materials also combined with GO as composites to prepare MMMs (not well introduced here neither).
I also suggest the authors only focus on the references published in the past 5 years. Not mean the old references are less valuable. The point is to better reflect the new trend and advanced in this area by citing latest papers.
Author Response
RESPONSE LETTER - Reviewer #2: membranes - 1940882
Thanks to the reviewers and editorial board that our manuscript entitled – “A potential Mixed matrix ultrafiltration membranes and their impact on the environment applications: A review” was suggested to be published after proper revision. We made the requested changes and motivated our choice and answers. Thank you for your time and effort and we hope that now this work can be accepted in this highly regarded journal.
Sincerely
Prof. Dr. Qusay F. Alsalhy
Membrane Technology Research Unit
Chemical Engineering Department
University of Technology,
Alsinaa Street No. 52
Baghdad, Iraq
Email: [email protected]; [email protected]
Reviewer #2:
Q1: This review article summarized the progress of MMMs with various NPs. The authors gave a quite detailed introduction about the NPs types, the effects of NPs on MMMs performances, and relevant applications. I admit this is enduring topic in membrane community. Dozens of similar reviews have been published in the past 5 years. My first concern is that what is the novelty of this review, comparing with these already published ones. I carefully checked the introduction section, but can’t found any valuable statements about the novelty of this paper.
Answer: Thank you for your comment; this review discusses the cutting edge research concerning the advancements of membranes modification employing various nanoparticles (NPs) like graphene oxide (GO), and the recent nanoparticles such as two-dimensional transition metal carbides and nitrides (MXenes), metal-organic framework works (MOFs), covalent organic framework (COFs) and other NPs and performance of MMMs for resolving environmental and energy-effective water purification
Based on the valuable remarks of the reviewer, we added the following paragraph in the end of the introduction section:
Mixed matrix membranes (MMMs) have attracted the attention of researchers worldwide, and the number of annual publications returned by the Google Scholar and ScienceDirect database has grown continuously doubled dozens of times from 2010 to 2022 and is still continuing to increase significantly. This indicates the great impact of MMMs on the performance of membranes which in turn prompted us to focus in this review on studying the effect of different NPs on the physicochemical properties, mechanical properties and performance for resolving environmental and energy-effective water purification. More specifically, this review focus on the effect of GO and hyper with other NPs as well as the recent nanoparticles such as two-dimensional transition metal carbides and nitrides (MXenes), metal-organic framework works (MOFs), covalent organic framework (COFs) due to their excellent properties on membrane characteristics to enhance removal efficiency of pollutants from wastewater.
Q2: Another concern is the NPs types mentioned in this paper. Most of them already well summarized by extensive review articles. Please simplify the 2.1, 5, and 6 sections. Because the relevant studies are not new already.
I suggest to adding some discussions about the newly emerging nanomaterials, such as MOFs, COFs, and MXene. These materials have been used as fillers for MMMs, but even limited introduced in this paper. Moreover, these materials also combined with GO as composites to prepare MMMs (not well introduced here neither).
Answer: We agree with this comment. It's already well summarized separately in many review articles that’s why we collect them in one review so that will help researchers in the future to obtain sufficient information about membrane for the updating of research for the last 10 years.
The incorporation of GO sheets into the polymer appears to make it highly attractive for multiple purposes due to its remarkable properties. These include a two-dimensional structure, the ability to promote negative surface charges, outstanding electron transport, high surface area innocuity, and remarkable chemical stabilities. GO can also change the roughness and mechanical properties of membranes and the effects of membrane fouling. It can increase water permeation flux and surface hydrophilicity, improve mechanical strength, reduce both organic and bio-fouling propensity, and efficiently separate pollutants.
Q3: I also suggest the authors only focus on the references published in the past 5 years. Not mean the old references are less valuable. The point is to better reflect the new trend and advanced in this area by citing latest papers.
Answer: Thank you for your suggestion; regarding the above mentioned critical remarks a new chapter(s) has been added to the manuscript as shown in section 6 (recent nanoparticles in the last few years) and section 8 (conclusion and future prospects).
In Introduction section:
"In recent few years, researchers have created 2D materials like graphene/graphene oxide (GO), covalent organic frameworks (COFs), and metal-organic frameworks (MOFs). Besides these materials, the 2D MXene material had also attracted the attention of researchers because it was only recently added for membrane fabrication and has unique chemical properties [57]. Because of their extendable layered structure as well as remarkable mass transfer channel, two-dimensional (2D) materials have recently demonstrated broad application prospects for the fabrication of high-performance membranes [58] [59]. Recent research has shown that inorganic MXene materials not only can enhance antifouling of the membranes but also improve removal efficiency [60]"
In section 2.1:
- MXene
In-situ reduction technology was used to create Ni@MXene magnetic particles, which were then external magnetic field attached to the top surface of the PES membrane. Huang et al, found the CR solution and colored emulsion demonstrated excellent decolonization ability. Furthermore, the antifouling mechanism might be demonstrated by the fact that the interaction between the Ni@MXene membrane and pollutants were relatively resistant than pure membrane [60]. While Shen et al, prepared the membrane by embedded MXene inside the membrane throughout the phase inversion by diffusing them in the coagulation bath. When compared to the pure PSF membrane, all composite polymeric membranes demonstrated significant improvements in water flux as well as BSA rejection. The improved antifouling feature is due to enhanced surface smoothness, increased hydrophilicity, and the MXene nanosheets' more negative zeta potential. Because of these significant enhancements [74].
- MOFs
Ma et al, found Applying MOF@GO composite as fillers to ultrafiltration membranes is a highly effective and promising technique for producing advanced water purification membranes. The hydrophilicity as well as water purification efficiency of UiO-66@GO/PES membranes were significantly improved after combining UiO-66@GO into PES membrane matrix. The pure water flux increased by 351% when compared to the pristine PES membrane, as did the rejection. The antifouling measurements have excellent antifouling performance [75].
- COFs
Zhang et al, improve that loading COFs materials under the best conditions could effectively have an efficient results for dye separation. Furthermore, the composed COF-based membrane passed the testing (which lasted 30 hours) and proved that it has great long-term stability and is also extremely dependable under highly acid/base conditions. As a result, the synthesized membrane seems to have the advantages of large-scale production [76].
- New nanoparticles in last few years
A large family of 2D materials such as 'MXene' has recently been the focus on by researchers due to their hydrophilicity, good chemical and structural stability, and high electrical conductivity. MXene materials are compete GOs in high stability as well as thermal resistance. Due to the number of layers of the MXene, they have been widely used in the manufacturing of super capacitors, fuel cell [108], lithium-ion batteries [109], and heavy metal adsorption [110][111] More than 30 MXene compositions have been experimentally synthesized due to the variety of M elements and three types of atomic structures While dozens of technically possible combinations have been predicted, a family of 2D materials is rapidly expanding [112]. MXene have been effectively incorporated with a wide range of materials including polymers in various field such as EMI shielding [113], energy storage devices [114], used also as electrode materials [115] and even filtration process. Han et al was prepared successfully 2D MXene/PES composite ultrafiltration membrane, he was improve when the MXene content increases, and so does the membrane's rejection of dyes and inorganic salts. So when MXene is tested at 0.2 g, the membrane has a high flux (above 115 L /m2.h) at 0.1 MPa. The membrane rejects Gentian violet (80.3%) and Congo red dye (92.3%) . The removal to inorganic salts is less than 23% with flux above (432 L /m2. h) at 0.1 MPa, denoting that membrane can be used in dye desalination as well as wastewater treatment with extreme efficiency [111].
MXene got a lot of attention for its ability to remove heavy metal ions from the wastewater such as Cu+2, Cd+2, and Cr+6. Yang et al. used different concentrations of Fe3O4NPs in the 2D MXene lamellar structure to create composite NF membranes. Where he could reach removal ratios in solution 63.2% for Cu+2, 64.1% for Cd+2, and 70.2% for Cr+6 [110]. One of the modification methods is the composite of nanomaterial's for membrane fabrication. Multiple nanomaterial's with hydrophilic groups were added into the membranes to improve the hydrophilicity. Huang et al study the separation and anti-fouling ability, were prepared TiO2@MXene composite and introduce into PES polymeric membrane. Were the higher flux he got of 1 g/l BSA solution (756.8 L /m2.h) and the rejection reach 70% [116]. MXene used also for removal of oil and dyes from wastewater as Ajibade et al studded by prepared MXene/O-MWCNT@PAN mixed matrix membrane for ultrafiltration membrane. The engagement of O-MWCNT within MXene nanosheets inside the membrane lead to quick passing of water molecules, while prevent oil droplets. Throughout the operational period, the rejection rates for oil and dye were about 97% and 99%, respectively. This describes the anti-swelling characteristic of the composite membrane's 3D MXene/O-MWCNT nanoparticles [117].
Many attempts had been made to create functional membranes for pollutant by combining new materials that have high-porosity have previously been studied for pollutant removal with ultrafiltration membranes [118]. MOFs that have composed of metallic ions and organic ligands have received a great deal of attention in the last decade due to their exceptional high surface area and porosity, functionalization capability, affinity for specific molecules, tunable chemical composition, and flexible structure. In comparison to traditional inorganic materials with 'rigid' structures, the organic nature of the MOF better compatibility with soft polymer matrices. These benefits may result in MOF/polymer blended membranes with increased permeability and stable solute rejection [119][120][121]. Sun et al. developing effective strategies MOF nanomaterial's for water purification membranes, were first synthesized from UiO-66-NH2 using a post-synthetic strategy. It was then combined with PSf to create hybrid UF membranes using a phase-inversion method. PSBMA coating formed round the MOF nanoparticles can give super hydrophilicity on UiO-66-PSBMA. Besides that, PSBMA coating in the distribution of UiO-66-PSBMA inside the PSf matrix, making a contribution to improved MOF Nano crystal as well as polymer phase compatibility. which The pure water flux increases and reaches its maximum (602 L m-2 h-1) at an UiO-66-PSBMA ( 0.3wt%) , which is 2.5 times than the pristine membrane (240 L m-2 h-1), and the BSA rejection at a relatively high level (> 98%) with porosity 73.4% and contact angle 66.3°. These results show that UiO-66-NH2 and PSBMA have synergic effect, and that both play an important role in enhancing the hybrid membrane's performance. The presence of fantastic hydrophilic MOF nanoparticle in the skin layer increases porous structure and water permeability. In the hybrid membranes, the macrovoids are repressed, as well as the finger-like pores operate through the cross-sectional structure and increased porosity [119].
COFs had also recently received a lot of attention in the fields of energy storage, catalysts, and separation. COFs, a new generation of crystalline natural porous material, are consisted of H, N, C, O, B, as well as other light atoms and have unique properties including such inevitable porous structure, good porous aperture, and active functional groups [122]. And it also a promising materials for membrane modification in treating wastewater and heavy metal recovery processes as Xu et al. studded the COFs/PVDF for lead removal [118]. Wang et al indicating that functionalities COFs membrane has great dye separation performance [123]. Pan et al. studded dye separation with high separation performance (99% for Methyl blue, Congo red, and Alcian blue and 93.91% for Orange GII). By a simple direct growth method for depositing continuous imine-based COF layers (TpPa-1, synthesized by Schiff-base reactions of 1,3,5-triformylphloroglucinol (Tp) with p-phenylenediamine (Pa-1)) has been developed. Aldehyde groups were first immobilized on a porous polyacrylonitrile (PAN) substrate as nucleation sites to encourage the development of a continuous TpPa-1 layer via good COF-to-substrate adhesion [124].
- Conclusion and future Prospects
- Because of the efficiencies seen and the advantages of MOFs, COFs, MXene nanoparticles, researchers give a lot of attention mixed them with polymeric membranes and these research has been grown rapidly over the last few years.
- MXene/polymer membranes have been summarized and have shown great promise in the manufacturing of MXene/polymer membranes, with overall performance superior to neat polymer films.
- COF materials with ordered channels and functionalized groups inside the channels provided a new strategy for fabricating high performance in the advancement of membrane processes for separation.
- Metal-Organic frameworks (MOFs) compared to the ordinary inorganic particles with 'rigid' frameworks, the unique nature of the MOF framework may support the growing with the polymer, allowing for good compatibility.
Reviewer 3 Report
The paper reviews mixed matrix ultrafiltration membranes. The topic is too general, and the focus of the work is not clear. The work is NOT scientifically strong, original, and organized. There are many works in the literature that reviewed some aspects of mixed matrix membranes to some extent. The contribution this work provides to science is not, therefore, clear. Some more specific comments:
- Title has a problem, "A potential mixed matrix membrane"?
- There are many of those figures related to the number of publications on a topic that doesn't provide any insight to readers. There is no reason for such overemphasis!
- Fig 1 on the types of membrane fouling is vague. There are way better quality figures showing various types of fouling mechanisms in the literature. Also, the scientific parts discussing different mechanisms are loosely written.
- There are many of those microscopy images that do not provide any valuable information related to this topic.
- There are many review papers and even books and book chapters about the mixed matrix or nanocomposite membranes. The work is not novel.
- Attractive figures with high quality must be added. The popular appeal is an important element of review works. Some related figures can be merged into one figure, and authors can also some changes to some of them with permission. The authors can also have their own figures, like comparative plots, schematics, etc., to make it more original.
- The materials provided in a review paper cannot be just common materials that you can find in many books and book chapters. The authors may want to ask themselves, what exactly are they contributing here to science?
- A critical literature review should provide some ideas about the gaps in the literature and some directions for future research that would help the researchers in the field.
Author Response
RESPONSE LETTER - Reviewer #3: membranes - 1940882
Thanks to the reviewers and editorial board that our manuscript entitled – “A potential Mixed matrix ultrafiltration membranes and their impact on the environment applications: A review” was suggested to be published after proper revision. We made the requested changes and motivated our choice and answers. Thank you for your time and effort and we hope that now this work can be accepted in this highly regarded journal.
Sincerely
Prof. Dr. Qusay F. Alsalhy
Membrane Technology Research Unit
Chemical Engineering Department
University of Technology,
Alsinaa Street No. 52
Baghdad, Iraq
Email: [email protected]; [email protected]
Reviewer #3: The paper reviews mixed matrix ultrafiltration membranes. The topic is too general, and the focus of the work is not clear. The work is NOT scientifically strong, original, and organized. There are many works in the literature that reviewed some aspects of mixed matrix membranes to some extent. The contribution this work provides to science is not, therefore, clear. Some more specific comments:
Q1: Title has a problem, “A potential mixed matrix membrane”?
Answer: Thank you for the valuable comment of the reviewer. Based on the above mentioned point, we modified the title of the manuscript – “Recent advances in membrane modification using nanoparticles and their impact on environmental applications: a review”.
Q2: There are many of those figures related to the number of publications on a topic that doesn't provide any insight to readers. There is no reason for such overemphasis!
Answer: Thank you for the useful comment. It was kept in mind, and the paper has been modified properly. All the modification has been highlighted by red in the manuscript.
Q3: Fig 1 on the types of membrane fouling is vague. There are way better quality figures showing various types of fouling mechanisms in the literature. Also, the scientific parts discussing different mechanisms are loosely written.
Answer: We fully agree with the reviewer’s comment, consequently the Fig 1 has been updated as follows:
Figure 1. Fouling mechanisms
Q4: There are many of those microscopy images that do not provide any valuable information related to this topic.
Answer: Thank you for the reviewer’s suggestion. We agree with this remark and the Fig. 15 has been removed from the article.
Q5: There are many review papers and even books and book chapters about the mixed matrix or nanocomposite membranes. The work is not novel.
Answer: We say thank you for this remark. As the reviewer mentioned there are many review papers have already summarized separately the advantages of different type of composite membranes. Although, we believe that collecting them in one new review paper and showing the extended field of current applications of these novel membranes can help the research community to obtain sufficient information about these processes, thereby opening new pathways in the membrane-based water and environmental cleaning technologies. To confirm the statement above, we would like to show a new a paragraph of the manuscript (see below). All the modification has been highlighted by red in the manuscript.
“The incorporation of GO sheets into the polymer appears to make it highly attractive for multiple purposes due to its remarkable properties. These include a two-dimensional structure, the ability to promote negative surface charges, outstanding electron transport, high surface area innocuity, and remarkable chemical stabilities. GO can also change the roughness and mechanical properties of membranes and the effects of membrane fouling. It can increase water permeation flux and surface hydrophilicity, improve mechanical strength, reduce both organic and bio-fouling propensity, and efficiently separate pollutants.”
Q6: Attractive figures with high quality must be added. The popular appeal is an important element of review works. Some related figures can be merged into one figure, and authors can also some changes to some of them with permission. The authors can also have their own figures, like comparative plots, schematics, etc., to make it more original.
- The materials provided in a review paper cannot be just common materials that you can find in many books and book chapters. The authors may want to ask themselves, what exactly are they contributing here to science?
Answer: Thank you for the suggestion. Figures are merged as much as possible, and we have already taken the permission for using Fig 5. Furthermore, we have inserted own figures and tables e.g. Figures 1-7, 12, 13 and Tables 1-3 as it was suggested.
Q7: A critical literature review should provide some ideas about the gaps in the literature and some directions for future research that would help the researchers in the field.
We say thank you for this very important and useful comment. We fully agree with the reviewer’s opinion and we believe that after a proper revision, following all the reviewer’s requests and suggestions, the quality and the content of the manuscript has been improved significantly. All the modification has been highlighted by red in the manuscript.
Round 2
Reviewer 2 Report
My comments have been well addressed.
Author Response
Thank you so much
Best Regards
Reviewer 3 Report
There are still some major issues with the paper.
1. The classification of mixed matrix membranes based on the type of nanoparticles must be revised. Why is SiO2 mentioned under 'transition metal oxides', it is a mineral nanomaterial. This is more critical for MXene, MOF, and COF. The authors need to look at this book, and more importantly first chapter, to come up with a better classification: Nanocomposite Membranes for Water and Gas Separation
Better classification could be minerals (like SiO2 and zeolites), metal oxides (like CuO, Fe2O3, ITO, ATO), carbon-based nanomaterials (like GO, CNF, CNT), and organic and metal-organic frameworks (like MOF and COF). ITO and ATO are double-element oxide nanomaterials that are used for making nanocomposite membranes. These papers can be added:
Novel nanocomposite polyethersulfone-antimony tin oxide membrane with enhanced thermal, electrical and antifouling properties
Thermally resistant and electrically conductive PES/ITO nanocomposite membrane
For the carbon-based nanomaterials for making mixed matrix membranes, this paper might help: Carbon-based polymer nanocomposite membranes for oily wastewater treatment
2. Some figures are still unnecessary and not that related to the topic. For example, Figure 1. If the authors still want to mention different fouling mechanisms of MF/UF membranes, they have to somehow use the terminologies in reviewing the literature. Also, it is suggested to increase the depth of discussion by adding the related equations. This paper could help: Characterization of boiler blowdown water from steam-assisted gravity drainage and silica–organic coprecipitation during acidification and ultrafiltration
3. Still, the number of statistical bar charts about the number of publications related to different topics is high and not informative.
4. The major problem with this review paper is that it doesn't have a strong organization on the main part of the paper that the authors need to provide insight into the topic. For example, in the subsection 'Applications of GO-polymeric membrane', some random results are provided without a coherent story and mentioning some major findings in the literature. I urge the authors to look at some review papers in the literature to get some ideas.
5. In the literature review part, the authors just provided some SEM images and flux/rejection results. Note that popular appeal of figures is really important in review papers. The authors can reduce the number of non-informative SEM images, add some eye-catching chemical structure or fabrication process figures, merge different figures into one figure, and have their own signature.
6. Again, a concern about the organization of the paper, what is the difference between subsections 'Application of polymeric membranes with NPs in the environment' and 'Applications of GO-polymeric membrane'? Also, what do they mean by 'Application in the environment'?
7. The topic is broad and if the authors really need to mention all the papers related to mixed matrix UF membranes, they have to review more than 5,000 papers. It is better to just focus on some novel nanomaterials. Some emerging nanomaterials like graphene nanoribbons and cellulose nanocrystals can be mentioned: Development of advanced nanocomposite membranes using graphene nanoribbons and nanosheets for water treatment and Fabrication of antifouling and antibacterial polyethersulfone (PES)/cellulose nanocrystals (CNC) nanocomposite membranes and Robust Polymer Nanocomposite Membranes Incorporating Discrete TiO2 Nanotubes for Water Treatment and Development of nanocomposite membranes by biomimicking nanomaterials
8. Some statements in the manuscript don't make sense either grammatically or scientifically. For example, what do the authors mean by 'More porous membranes can be made by adding SiO2, SiC, Si3N4, and TiO2.' why do we need to make more porous membranes, and how? General statements that do not provide any valuable insight must be avoided. Almost all statements I need color on page 24 are too general and are not informative!
Author Response
Dear Editor
I would like to submit a revised manuscript entitled " Classification of nanomaterial's and the effect of Graphene oxide (GO) and the recent nanoparticles on the ultrafiltration membrane and their applications: a review " for consideration for publication in the Membranes. This article is revised according to the reviewers’ comments and highlighted in different color in the revised manuscript. Please answer to the reviewers’ comments as in the following report.
I appreciate the effort of the Editor and Reviewer to improve our article, thank you.
Your consideration for this manuscript with revised form is highly appreciated.
Sincerely
Prof. Dr. Qusay F. Alsalhy
Membrane Technology Research Unit
Chemical Engineering Department
University of Technology,
Alsinaa Street No. 52
Baghdad, Iraq
Email: [email protected]; [email protected]
Answer to the Reviewers Comments:
There are still some major issues with the paper.
- 1. The classification of mixed matrix membranes based on the type of nanoparticles must be revised. Why is SiO2 mentioned under 'transition metal oxides', it is a mineral nanomaterial. This is more critical for MXene, MOF, and COF. The authors need to look at this book, and more importantly first chapter, to come up with a better classification: Nanocomposite Membranes for Water and Gas Separation
Better classification could be minerals (like SiO2 and zeolites), metal oxides (like CuO, Fe2O3, ITO, ATO), carbon-based nanomaterials (like GO, CNF, CNT), and organic and metal-organic frameworks (like MOF and COF). ITO and ATO are double-element oxide nanomaterials that are used for making nanocomposite membranes. These papers can be added:
Novel nanocomposite polyethersulfone-antimony tin oxide membrane with enhanced thermal, electrical and antifouling properties
Thermally resistant and electrically conductive PES/ITO nanocomposite membrane
For the carbon-based nanomaterials for making mixed matrix membranes, this paper might help: Carbon-based polymer nanocomposite membranes for oily wastewater treatment
Ans:
Thanks for your suggestions the classification have been updated depending on the type of nanomaterial's. Thanks for the researches they help to much, they have been added to the review too.
- Introduction
- Mixed matrix membranes (MMM)
2.1. Classification of nanomaterial's
2.1.1 Mineral Nanomaterial (SiO2 and zeolite),
2.1.2 metals oxide (CuO, ZrO2), ZnO, ATO, Fe2O3 and WOX ) ,
2.1.3 Two-dimensional transition
2.1.4 Metal-organic framework (MOFs)
2.1.5 Covalent organic frameworks (COFs),
2.1.6 Carbon-based nanomaterial's (Graphene oxide (GO) and CNT, CNF)
- Effect of NPs on the morphology of the PPSU, PES, PVC membranes
- Application of MMMs
- Conclusion and future Prospects
- 2. Some figures are still unnecessary and not that related to the topic. For example, Figure 1. If the authors still want to mention different fouling mechanisms of MF/UF membranes, they have to somehow use the terminologies in reviewing the literature. Also, it is suggested to increase the depth of discussion by adding the related equations. This paper could help: Characterization of boiler blowdown water from steam-assisted gravity drainage and silica–organic coprecipitation during acidification and ultrafiltration
Ans:
The part about fouling mechanisms updated and we have included the equations of the fouling mechanisms. please go to see page 2 and 3. Again thanks for the research, it have been included in the review.
- 3. Still, the number of statistical bar charts about the number of publications related to different topics is high and not informative.
Ans:
Figures about "publication of some nanomaterial's" have been deleted.
And there is an explanations about the other statistical bar charts please check page 13 about figure 3. And page 20 about figures 4,5.
- 4. The major problem with this review paper is that it doesn't have a strong organization on the main part of the paper that the authors need to provide insight into the topic. For example, in the subsection 'Applications of GO-polymeric membrane', some random results are provided without a coherent story and mentioning some major findings in the literature. I urge the authors to look at some review papers in the literature to get some ideas.
Ans:
We have organized the paper again and take the benefits parts from 'Applications of GO-polymeric membrane' and have been added to the ' Application of the MMMs' please go to see section 4 page 27.
- 5. In the literature review part, the authors just provided some SEM images and flux/rejection results. Note that popular appeal of figures is really important in review papers. The authors can reduce the number of non-informative SEM images, add some eye-catching chemical structure or fabrication process figures, merge different figures into one figure, and have their own signature.
Ans:
Thanks for your suggestions we have reduce the number of SEM images and flux/rejection results please go to see section 3 pages (24,25 and 26). And section 4 page 28.
- 6. Again, a concern about the organization of the paper, what is the difference between subsections 'Application of polymeric membranes with NPs in the environment' and 'Applications of GO-polymeric membrane'? Also, what do they mean by 'Application in the environment'?
Ans:
Again, the organization of the paper has been updated. and take the benefits parts from 'Applications of GO-polymeric membrane' and have been added to the ' Application of the MMMs' please go to see section 4 page 27.
Update 'Application of polymeric membrane in the environment' to " Application of MMMs"
- 7. The topic is broad and if the authors really need to mention all the papers related to mixed matrix UF membranes, they have to review more than 5,000 papers. It is better to just focus on some novel nanomaterials. Some emerging nanomaterials like graphene nanoribbons and cellulose nanocrystals can be mentioned: Development of advanced nanocomposite membranes using graphene nanoribbons and nanosheets for water treatment and Fabrication of antifouling and antibacterial polyethersulfone (PES)/cellulose nanocrystals (CNC) nanocomposite membranes and Robust Polymer Nanocomposite Membranes Incorporating Discrete TiO2Nanotubes for Water Treatment and Development of nanocomposite membranes by biomimicking nanomaterials
Ans:
The title " Classification of nanomaterial's and the effect of Graphene oxide (GO) and recent nanomaterial's on the ultrafiltration membrane and their applications: a review "
The title has been changed depend on the content of the review. We have focus on Graphene oxide specifically and some recent nanomaterial's and their effect on the polymeric membrane and their applications.
- 8. Some statements in the manuscript don't make sense either grammatically or scientifically. For example, what do the authors mean by 'More porous membranes can be made by adding SiO2, SiC, Si3N4, and TiO2.' why do we need to make more porous membranes, and how? General statements that do not provide any valuable insight must be avoided. Almost all statements I need color on page 24 are too general and are not informative!
Ans:
This statement was deleted 'More porous membranes can be made by adding SiO2, SiC, Si3N4, and TiO2.'
We have taken the benefit parts from page 24( recent nanomaterial's) and added to the sections 2, "Classification of nanomaterial's" and delete the rest.
Round 3
Reviewer 3 Report
The comments are applied properly.